# Human RAD51 paralogue SWSAP1 fosters RAD51 filament by regulating the anti-recombinase FIGNL1 AAA+ ATPase

Kenichiro Matsuzaki[1], Shizuka Kondo[1,2], Tatsuya Ishikawa[1,2] & Akira Shinohara [1]

RAD51 assembly on single-stranded (ss)DNAs is a crucial step in the homology-dependent repair of DNA damage for genomic stability. The formation of the RAD51 filament is promoted by various RAD51-interacting proteins including RAD51 paralogues. However, the mechanisms underlying the differential control of RAD51-filament dynamics by these factors remain largely unknown. Here, we report a role for the human RAD51 paralogue, SWSAP1, as a novel regulator of RAD51 assembly. *Swsap1*-deficient cells show defects in DNA damage-induced RAD51 assembly during both mitosis and meiosis. Defective RAD51 assembly in SWSAP1-depleted cells is suppressed by the depletion of FIGNL1, which binds to RAD51 as well as SWSAP1. Purified FIGNL1 promotes the dissociation of RAD51 from ssDNAs. The dismantling activity of FIGNL1 does not require its ATPase but depends on RAD51-binding. Purified SWSAP1 inhibits the RAD51-dismantling activity of FIGNL1. Taken together, our data suggest that SWSAP1 protects RAD51 filaments by antagonizing the anti-recombinase, FIGNL1.

[1] Laboratory of Genome and Chromosome Functions, Institute for Protein Research, Osaka University, 3-2 Yamadaoka, Suita, Osaka 565-0871, Japan. [2] Graduate School of Science, Department of Biological Sciences, Osaka University, 3-2 Yamadaoka, Suita, Osaka 565-0871, Japan. Correspondence and requests for materials should be addressed to K.M. (email: k.matsuzaki@protein.osaka-u.ac.jp) or to A.S. (email: ashino@protein.osaka-u.ac.jp)

Homologous recombination (HR) is essential to maintain genome stability by repairing DNA double-strand breaks (DSBs) and stalled DNA replication forks. In DSB repair, HR requires the formation of single-stranded DNA (ssDNA) through the processing of the DSB ends. Subsequently, the ssDNA is used to find a homologous double-stranded DNA (dsDNA) and invaded it to form a displacement loop called a D-loop. These homology search and strand invasion steps are catalyzed by a RAD51/RecA family protein[1–3]. In eukaryotes, RAD51 binds to ssDNA in the presence of ATP to form a right-handed helical-filament on the DNA[4–7]. The RAD51 filament is a key protein machinery for the homology search and strand exchange.

RAD51 filaments are highly dynamic. Defective assembly or improper assembly of RAD51 filaments leads to genomic instability. Therefore, RAD51-filament formation is tightly regulated both temporally and spatially by several positive and negative factors. Once ssDNA is formed in vivo, the ssDNA binding protein, replication protein-A (RPA), tightly binds to the ssDNA, which inhibits the loading of RAD51. To promote RAD51 loading on RPA-coated ssDNA, RAD51 requires positive regulators called RAD51 mediators that act to load RAD51 on RPA-bound DNA. RAD51 mediators, which bind directly to RAD51, include the breast cancer susceptibility gene, BRCA2, as well as RAD52 and RAD51 paralogues[8].

RAD51 paralogues share 20–30% amino acid sequence similarity with RAD51[9,10]. Each organism bears unique sets of RAD51 paralogues: RAD55 and RAD57 in the budding yeast and RAD51B, RAD51C, RAD51D, XRCC2, XRCC3, and SWSAP1 in mammals. As a sub-family of the RAD51 paralogue, recent studies on budding yeast Psy3-Csm2-Shu1-Shu2 (a.k.a. Shu complex) identified Psy3-Csm2-Shu1 as a structural homolog of Rad51 without any sequence similarity[11,12]. In humans, RAD51B, RAD51C, RAD51D, XRCC2, and XRCC3 form two distinct complexes, RAD51B-C-D-XRCC2 (BCDX2), and RAD51C-XRCC3 (CX3)[13]. SWSAP1 was first identified as an interacting protein with SWS1, a human homolog of fission yeast Sws1 and budding yeast Shu2[11,14,15]. Although all human RAD51 paralogues are required for proper RAD51 assembly in vivo[16–18], the mechanisms underlying RAD51-filament formation as well as the role that each paralogue plays in RAD51 assembly remain poorly defined. Several studies have reported that RAD51 paralogue knockout (KO) mice exhibit embryonic lethality, similar to RAD51 KO mice[8,19–23], making the assessment of the in vivo functions of the RAD51 paralogues difficult.

RAD51 disassembly is also controlled by various proteins. The negative regulators of RAD51 assembly include a RAD51-filament remodeller. In budding yeast, Srs2 DNA helicase has the ability to dismantle Rad51 filaments[24]. Similar to yeast Srs2, human BLM, RECQL5, FBH1, and PARI have negatively regulate HR by disrupting the RAD51 filament[25–28]. These are defined as anti-recombinases. Although disruption of the RAD51 filament is necessary for genomic stability to prevent inappropriate HR, these anti-recombinases should be inert on active RAD51 filaments at a normal HR site. However, this highly dynamic control of the assembly/disassembly of RAD51 filament by multiple positive and negative regulators is poorly understood.

In this study, we report a novel function of the RAD51 paralogue, SWSAP1, in RAD51 assembly. Human SWSAP1 interacts with RAD51 through its conserved Phe-X-X-Ala (FxxA) BRC-like motif and this interaction is required for DNA damage-induced RAD51-focus formation. We also find that SWSAP1 interacts with FIGNL1 (fidgetin-like 1), an AAA+ ATPase involved in HR[29]. We have shown that FIGNL1 depletion suppresses RAD51-focus formation defects in SWSAP1-deficient cells, indicating that FIGNL1 facilitates RAD51 disassembly in the

absence of SWSAP1. Notably, we show that purified FIGNL1 promotes the dissociation of RAD51 from ssDNA in an ATPase activity-independent manner. Purified SWSAP1 antagonizes the RAD51-filament dismantling activity of FIGNL1 in vitro. Based on our observations, we propose that SWSAP1 stabilizes the RAD51 filament by counteracting the FIGNL1 anti-recombinase. Our findings reveal a unique stabilization mechanism of RAD51 filaments by a human RAD51 paralogue.

## Results

**FxxA motif in SWSAP1 is crucial for RAD51-focus formation.** To investigate the molecular mechanism by which SWSAP1 promotes RAD51 assembly, we examined the physical interaction between SWSAP1 and RAD51. FLAG-SWSAP1 was co-expressed with Myc-RAD51 in 293T cells and their association was confirmed by co-immunoprecipitation (co-IP) (Fig. 1a). To determine the region in SWSAP1 that binds to RAD51, we generated various truncated versions of SWSAP1 (Fig. 1b). The two middle regions (57–110 aa and 109–169aa) of SWSAP1 are not important for the interaction with RAD51. On the other hand, a C-terminus (169–229 aa) deletion reduced the interaction and an N-terminus (1–57 aa) deletion almost abolished binding (Fig. 1d). We observed that the SWSAP1 N-terminus contains the FxxA motif, a highly conserved RAD51-binding motif[30], which was originally identified in BRC repeats of the BRCA2 protein, referred as to BRC variant (BRCv) (Fig. 1c). We mutated phenylalanine 23 and alanine 26 on the FxxA motif to glutamate (F23E, A26E; SWSAP1-EE mutant). We also mutated a conserved 44–49aa (PLQSMP) region downstream to the FxxA motif by substituting all 6 residues with alanine in the SWSAP1-A3 mutant (Fig. 1c). SWSAP1-EE showed reduced binding to RAD51 and the binding of the SWSAP1-A3 mutant to RAD51 in vivo was greatly abrogated relative to the wild-type SWSAP1 (Fig. 1e). These results indicate that the FxxA motif and its downstream 44–49aa region on SWSAP1 are crucial for RAD51 interaction although we cannot exclude the possibility that these amino acid changes alter proper protein folding and activity.

To assess the effect of the SWSAP1–RAD51 interaction on RAD51 assembly, we monitored DNA damage-associated RAD51-focus formation in SWSAP1 mutants by immunofluorescence microscopy. Similar to previous observations[14], SWSAP1 knockdown decreased the punctate staining of RAD51 foci, which was induced by the treatment of U2OS cells with camptothecin (CPT), a topoisomerase-I inhibitor (Fig. 1f). The expression of siRNA-resistant wild-type SWSAP1 suppressed the defect in RAD51-focus formation in the SWSAP1-depleted cells. However, this defect was not rescued in the SWSAP1-EE mutant (Fig. 1f). Hence, the interaction of SWSAP1 and RAD51 through the FxxA motif of SWSAP1 is required for DNA damage-induced RAD51-focus formation.

**SWSAP1 interacts with FIGNL1.** To gain insight into the function of SWSAP1 in HR, we looked for an interacting partner of SWSAP1 and found that SWSAP1 interacts with Fidgetin-like 1 (FIGNL1) protein. FIGNL1 is an AAA+ ATPase protein that is known to bind to RAD51 and has a role in HR[29]. FLAG-SWSAP1 and Myc-FIGNL1 were co-expressed in 293T cells and subjected to co-IP using an anti-Myc antibody. FLAG-SWSAP1 was recovered in the Myc-FIGNL1 immunoprecipitate from cells that expressed both proteins (Fig. 2a). This result is supported by the previous observation that human SWSAP1 peptides (C19orf39) were identified by mass spectroscopy as FIGNL1-binding proteins[29]. The interaction between SWSAP1 and FIGNL1 was also confirmed by using recombinant proteins. A His-tagged SWSAP1–SWS1 complex and GST-tagged FIGNL1ΔN, in which N-terminal 284aa

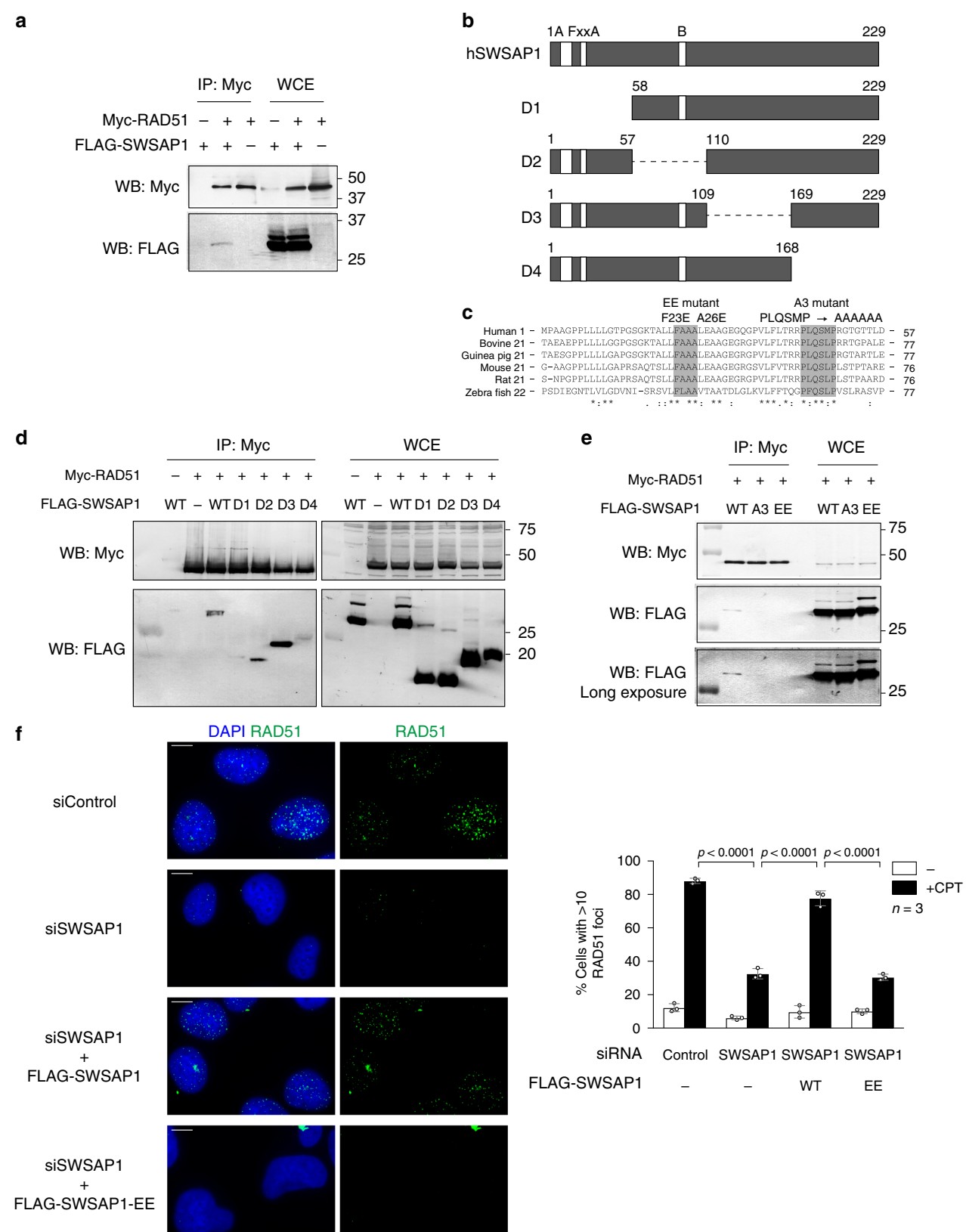

were deleted (Supplementary Fig. 1a), were purified from bacteria and were used for a GST pulldown assay. The SWSAP1–SWS1 complex bound to GST–FIGNL1ΔN fusion protein, but not GST alone (Fig. 2b). This result indicates that SWSAP1 directly interacts with FIGNL1 rather than a mediation by RAD51.

To identify a FIGNL1-binding site in SWSAP1, we used a series of deletion mutants of SWSAP1 (Fig. 1b, c). Co-IP revealed that the N-terminus (1–57 aa) and C-terminus (169–229 aa) of SWSAP1 are necessary for its interaction with FIGNL1 (Fig. 2c). Since the N-terminus of SWSAP1 is also important for binding to

**Fig. 1** SWSAP1–RAD51 interaction is required for RAD51-focus formation. **a** Immunoblots of Myc-RAD51 immunoprecipitates. Myc-RAD51 and FLAG-SWSAP1 were expressed in 293T cells and immunoprecipitated with anti-Myc beads after 72 h transfection. The immunoprecipitates (IP) and whole-cell extract (WCE) were probed with indicated antibodies. **b** Schematic representation of human SWSAP1 truncation mutants used in Figs. 1d and 2c. **c** Amino acid sequence comparison of conserved SWSAP1 N-terminal region. Amino acid substitutions are indicated above the sequences. Substitution of amino acids is shown on top. **d**, **e** Co-immunoprecipitation analysis of SWSAP1 mutants. Myc-RAD51 and various FLAG-SWSAP1 mutants shown in **b** were analyzed by IP as shown in **a**. **f** Left, Representative images of immuno-staining analysis of RAD51 focus in control U2OS cells and SWSAP1-depleted cells (top two raw) at 22 h after the treatment of 100 nM of camptothecin (CPT). siRNA-resistant SWSAP1-expressing cells and siRNA-resistant SWSAP1-EE-expressing cells lines after SWSAP1 depletion were analyzed for RAD51 staining (bottom two raw). Right, Quantification of RAD51 focus-positive cells (more than 10 foci per a cell) in the indicated cell lines. At each point, more than 200 cells were counted. Bar 10 μm. Data are mean ± s.d. ($n = 3$, three biological independent). Statistical significance was measured by Mann–Whitney's $U$-test. Statistics and reproducibility; see accompanying Source Data

RAD51, we examined the role of the FxxA motif of SWSAP1 in the interaction with FIGNL1. Co-IP assays revealed that SWSAP1-EE binds to FIGNL1 at a comparable level as the wild-type SWSAP1 (Fig. 2c), indicating that SWSAP1 uses a different interface for binding to FIGNL1 from for binding to RAD51. To narrow down the interaction region in SWSAP1, we generated two C-terminal truncation mutants, D4N and D4C, in which residues 169–193 and 194–229 were deleted, respectively (Fig. 2d). Reduced interaction of FIGNL1 with SWSAP1-D4C was observed, but not with the D4N mutant (Fig. 2e). Various substitutions of highly conserved amino acids in the C-terminal 35aa region of SWSAP1 revealed that the single amino acid substitution of lysine 221 to arginine impaired the interaction of SWSAP1 with FIGNL1 (Fig. 2f and Supplementary Fig. 2a–c).

In order to elucidate the role of the SWSAP1–FIGNL1 interaction in RAD51 assembly, we examined DNA damage-induced RAD51-focus formation. Reduced RAD51-focus formation in SWSAP1-depleted cells was suppressed by the expression of siRNA-resistant wild-type SWSAP1 but not by SWSAP1–K221R expression (Fig. 2g), suggesting that the SWSAP1–FIGNL1 interaction is critical for RAD51 assembly.

**FIGNL1 suppresses RAD51-focus formation in SWSAP1-depleted cells**. Next, we investigated the functional relationship between SWSAP1 and FIGNL1 in RAD51 assembly by depleting SWSAP1 and/or FIGNL1. Consistent with a previous study[29], CPT-induced RAD51 foci appear normal in FIGNL1-depleted cells. Unexpectedly, the depletion of FIGNL1 suppressed defective RAD51-focus formation in SWSAP1-depleted cells. RAD51-focus formation in SWSAP1/FIGNL1-double depleted cells was similar to that in control and FIGNL1-depleted cells (Fig. 3a, b). We also verified that FIGNL1 depletion suppresses a defect conferred by the depletion of SWSAP1-binding partner, SWS1[14,15]. Indeed, impaired RAD51-focus formation induced by SWS1 depletion was also suppressed by FIGNL1 knockdown (Supplementary Fig. 3). These data suggest that FIGNL1 could disrupt RAD51 foci, possibly as an anti-recombinase, in the absence of SWSAP1–SWS1.

In mammals, several anti-recombinases that inhibit HR have been described. These include BLM, PARI, and RTEL1[25,27]. We examined whether the depletion of other anti-recombinases suppresses the RAD51-assembly defect in SWSAP1-deficient cells. When BLM, PARI, or RTEL1 was depleted along with SWSAP1, reduced RAD51-focus formation in SWSAP1-depleted cells was not rescued (Fig. 3a, b). These results suggest that the functional interaction of SWSAP1 is specific to FIGNL1 and not BLM, PARI, or RTEL1.

SWSAP1 is one of the RAD51 paralogues required for efficient RAD51 assembly. We extended the relationship of SWSAP1 and FIGNL1 to RAD51C, a RAD51 paralogue and a common component of the BCDX2 and CX3 complexes[13]. As previously reported, RAD51C deficiency leads to defective RAD51-focus formation[16,17]. DNA damage-induced RAD51-focus formation in

RAD51C/FIGNL1-double depleted cells was almost similar to that in RAD51C-depleted cells, indicating that FIGNL1 depletion did not suppress the RAD51-assembly defect in RAD51C-deficient cells (Fig. 3c). These data suggest that FIGNL1 prevents RAD51 assembly in the absence of SWSAP1, but not in the absence of RAD51C.

Since FIGNL1 directly binds to the C-terminus of SWSAP1, we tested the role of this interaction of FIGNL1 in RAD51 assembly using the interaction-defective SWSAP1–K221R mutant. A marked increase in RAD51 assembly was observed by depleting FIGNL1 in the SWSAP1–K221R mutant, indicating that the SWSAP1–FIGNL1 interaction is required to inhibit the anti-recombinase activity of FIGNL1 (Fig. 2g).

**Spontaneous chromosome fragmentations in SWSAP1-depleted cells are suppressed by FIGNL1 depletion**. The restoration of RAD51-focus formation by FIGNL1 depletion in SWSAP1-depleted cells prompted us to test whether the restored RAD51 assembly possesses the ability to repair DNA damage. To this end, we examined chromosome instability on mitotic chromosome spreads in U2OS cells. One type of chromosome instability, spontaneous chromosome fragmentation, was analyzed in cells depleted of SWSAP1, FIGNL1, or both. A significant increase in chromosome fragmentation was observed in SWSAP1-depleted cells relative to control cells ($7.5 ± 2.1$(SD; standard deviation) % [$4.67 ± 1.35$ fragmentation per a spread] in control cells vs $10.1 ± 1.8$% [$6.15 ± 1.09$] in SWSAP1-depleted cells) (Fig. 3d, e), indicating that SWSAP1 is critical for maintaining chromosome stability. FIGNL1 depletion did not affect the frequency of fragmentation. The increase of the chromosome abnormality in SWSAP1-depleted cells was suppressed by the FIGNL1 depletion, implying that rescued RAD51 assembly in SWSAP1-depleted cells by the FIGNL1 depletion could lead to the repair of spontaneous DNA damage and protect cells from chromosome instability.

**FIGNL1 promotes RAD51 dissociation from DNAs**. Suppression of the RAD51-focus formation defect in SWSAP1-deficient cells by FIGNL1 depletion raised the possibility that FIGNL1 prevents RAD51-filament formation in the absence of SWSAP1. To elucidate the role of FIGNL1 in the RAD51-filament formation, we examined the binding of RAD51 to an 83nt ssDNA immobilized to magnetic beads (Fig. 4a). We pre-incubated RAD51 with the ssDNA on beads in the presence of ATP and $Mg^{2+}$. After the formation of the RAD51-ssDNA complex, the purified N-terminal-truncated human FIGNL1 protein (deletion of amino acids 1–284; FIGNL1ΔN) containing an ATPase domain with highly conserved Walker A/B motifs, RAD51-binding domain, and VPS4 domain[29] was added to the complex (Supplementary Fig. 1a, b). After incubation, we separated beads containing ssDNA-bound RAD51 from the supernatant containing released RAD51 from the DNA. In the presence of ATP,

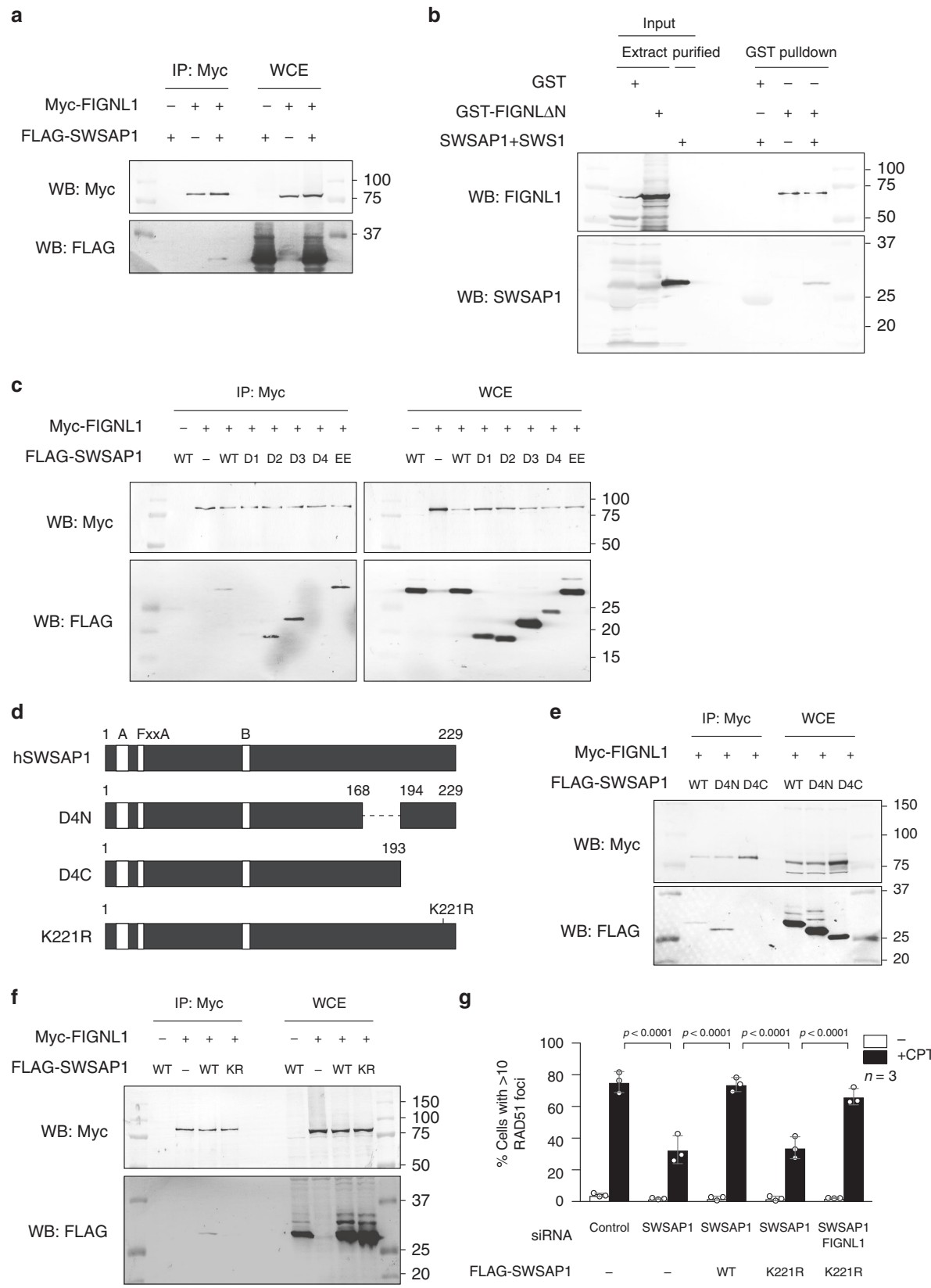

RAD51 stably binds to the ssDNA without any dissociation in supernatant. When different concentrations of FIGNL1 were added to pre-assembled RAD51-ssDNA, RAD51 was recovered in the supernatant fractions in proportion to the FIGNL1 concentration, indicating the dissociation of RAD51 from ssDNA. At

$0.5\ \mu M$ of FIGNL1, almost 31.2% of the RAD51 was recovered, however, in the absence of FIGNL1 only 1.0% was recovered (Fig. 4b). Concomitantly, the amounts of RAD51 bound to ssDNA (ssDNA-bound) were reduced. These data indicate that FIGNL1 promotes the dissociation of RAD51 from ssDNA. Our

**Fig. 2** SWSAP1–FIGNL1 interaction is required for RAD51-focus formation. **a** Immunoblots of Myc-FIGNL1 immunoprecipitates. Myc-FIGNL1 and FLAG-SWSAP1 were expressed in 293T cells and, after 72 h, cell lysates were immunoprecipitated with anti-Myc beads. IP and WCE were subjected to western blotting with indicated antibodies. **b** Direct interaction between SWSAP1 and FIGNL1. *E. coli* extracts expressing GST or GST–FIGNL1ΔN were incubated with glutathione beads. After wash, the beads were incubated with purified SWSAP1–SWS1 complex. After collection of the beads, samples were eluted with glutathione and subjected to western blotting. **c** Co-immunoprecipitation analysis of SWSAP1 mutants. Myc-FIGNL1 and indicated FLAG-SWSAP1 truncation mutants were expressed in 293T cells and, after 72 h transfection, extracts were used for IP and analyzed by western blotting with indicated antibodies. **d** Schematic representation of SWSAP1 C-terminal truncations and K221R mutants. **e**, **f** Co-immunoprecipitation analysis of SWSAP1 C-terminus mutants with FIGNL1. FLAG-SWSAP1 C-terminal truncation and FLAG-SWSAP1–K221R mutants were co-expressed with Myc-FIGNL1 in 293T cells and, after 72 h, transfection, WCE was used for IP using anti-Myc to detect the binding to Myc-FIGNL1. **g** Quantification of RAD51-positive cell in SWSAP1–K221R mutants. U2OS cells with siRNA transfection with or without expression of siRNA-resistant SWSAP1 or SWSAP1–K221R, the cells were treated with 100 nM of CPT for 22 h, RAD51-focus formation was analyzed as shown in Fig. 1f. Bar 10 μm. Data are mean ± s.d. ($n = 3$, three biological independent). Statistical significance was measured by Mann–Whitney's *U*-test. Statistics and reproducibility; see accompanying Source Data

in vivo and in vitro studies described here suggest that FIGNL1 is a new member of RAD51-filament remodeling proteins with anti-recombinase activity.

$Ca^{2+}$ ion is known to stabilize RAD51 filament[31,32]. We examined whether FIGNL1 is able to dissociate RAD51-ssDNA formed in the presence of ATP and $Ca^{2+}$ and found that FIGNL1 did not dissociate RAD51-ssDNA stabilized by $Ca^{2+}$ ion. This suggests that ATP hydrolysis by RAD51 might have a role in FIGNL1-mediated RAD51 remodeling (Supplementary Fig. 5a). We also examined the effect of different lengths of ssDNA. In addition to the 83 nt ssDNA, we formed RAD51 filament on 43nt and 153nt ssDNAs (same concentration in nucleotides) and examined the effect of FIGNL1 (Supplementary Fig. 5b). FIGNL1 dissociates RAD51 from 153nt ssDNA in a similar to RAD51 from 83nt ssDNA. On the other hand, RAD51 on 43nt ssDNA was a better substrate for the FIGNL1 activity relative 83nt and 153nt ssDNAs. This suggests that FINGL1 may dissociate RAD51 protomers from their ends. This finding is compatible with a solution-based sequestering model (Discussion).

To further elucidate the molecular mechanism by which FIGNL1 facilitates RAD51 dissociation from DNA, we generated a FIGNL1 Walker A mutant deficient in ATPase (K447R; FIGNL1ΔN-KR) (Supplementary Fig. 1a, c, h). Similar to wild-type FIGNL1 protein, the FIGNL1ΔN-K447R mutant protein shows normal RAD51-binding activity (Supplementary Fig. 1e), and promotes the dissociation of RAD51 from DNA with slight increase, indicating that the AAA+ ATPase activity of FIGNL1 is not necessary for RAD51-dismantling activity (Fig. 4c). Previously, it has been shown that FIGNL1 binds to RAD51 through the FIGNL1's RAD51-binding domain (FRBD) containing the FxxA BRCv motif. We mutated the FxxA motif (F295E, A298E; FIGNL1ΔN-EE) and examined purified FIGNL1ΔN-EE protein for RAD51-dismantling activity. Consistent with previous observations, the FIGNL1ΔN-EE mutation reduced binding to RAD51 (Supplementary Fig. 1e). FIGNL1ΔN-EE protein showed a 3.2-fold decrease in activity to dissociate RAD51 from ssDNA relative to wild-type protein (Fig. 4c). These results suggest that FIGNL1 disassembles RAD51 from DNA through its interaction with RAD51. We also measured ATP hydrolysis activity of wild-type FIGNL1ΔN, FIGNL1ΔN-KR and FIGNL1ΔN-EE proteins (Supplementary Fig. 1g, h). Both wild-type FIGNL1ΔN and FIGNL1ΔN-EE proteins hydrolyzed ATP. As expected, FIGNL1ΔN-KR proteins showed a large reduction of the ATPase activity.

To examine the effect of these mutations on in vivo FIGNL1 activity, we analyzed the DNA damage-induced RAD51-focus formation in siRNA-resistant wild-type FIGNL1-, FIGNL1-EE-, and FIGNL1-KR-expressing cells after double depletion of FIGNL1 and SWSAP1. The expression of wild-type FIGNL1 decreased RAD51-focus formation caused by the SWSAP1 depletion. However, FIGNL1-EE expressing cells did not show

marked reduction of the RAD51-focus formation relative to wild-type FIGNL1-expressing cells, suggesting that FIGNL1-EE has a compromised ability to suppress RAD51-focus formation in vivo (Supplementary Fig. 1i). Corresponding with results of in vitro RAD51 disassembly assay, RAD51-focus formation in FIGNL1-KR cells was slightly less than that in wild-type expressing cells, indicating the activity of FIGNL1-KR to suppress RAD51-focus formation is slightly higher than that of wild-type FIGNL1 (*P*-value = 0.39, Mann–Whitney's *U*-test; Supplementary Fig. 1i).

The above in vivo experiments showed that SWSAP1 protects RAD51 assembly from FIGNL1 (Fig. 3). To determine whether SWSAP1 protects RAD51 filaments from FIGNL1, we purified human SWSAP1 without its binding partner, SWS1, and examined the effect on RAD51-dismantling activity by FIGNL1. When SWSAP1 was added, most was subsequently recovered in the supernatant, indicating that the majority of SWSAP1 proteins do not bind to the ssDNA beads under these conditions. Thus, we pre-incubated SWSAP1 with FIGNL1 for 30 min and then added the mixture to RAD51-coated ssDNA beads (Fig. 4a). Only 21.2% of dissociated RAD51 was detected in the supernatant in the presence of SWSAP1, while 36.3% of RAD51 was found in the supernatant in the absence of SWSAP1 (Fig. 4d). This indicates that SWSAP1 inhibits the RAD51-filament dismantling activity of FIGNL1 when pre-incubated together.

**Swsap1-deficient mice are defective in gamete formation.** To investigate the function of *Swsap1* in vivo, we generated *Swsap1* KO mice by deleting exon 2, which encodes 82–278aa of the SWSAP1 protein (Fig. 5a). *Swsap1*^{−/−} mice were viable, showed normal growth and weight gain, and did not show any obvious developmental abnormalities (Supplementary Fig. 4a, b). However, *Swsap1*^{−/−} mice had ~1/3 smaller sizes of testis than did the wild-type and *Swsap1*^{+/−} mice (Fig. 5c and Supplementary Fig. 4c). To investigate the effect of *Swsap1*-deficiency on spermatogenesis, we examined the testes from *Swsap1*^{−/−} mice and their littermates. Histological analysis of wild-type and *Swsap1*-deficient testes revealed that almost all seminiferous tubules were found to be atrophic in *Swsap1*^{−/−} mice (Fig. 5d). Histological analysis of adult ovary from the *Swsap1*^{+/−} mice showed lack of developing follicles (Fig. 5e, Supplementary Fig. 4d), indicating that *Swsap1* is essential for female meiosis. This implies that *Swsap1*-deficiency results in defective spermatogenesis and oogenesis, consistent with the phenotypes of *Swsap1* mutant mice with a frame-shift indel mutation generated by the TALEN nuclease[33].

In human cells, SWSAP1 stabilizes RAD51 filaments (see above). To determine whether *Swsap1* is required for RAD51 assembly during meiosis, leptotene spermatocytes, which were defined as a cell containing un-synapsed SYCP3 axes, were examined for RAD51-focus formation on nuclear spreads. A marked reduction in the number of RAD51 foci per cell was observed in *Swsap1*^{−/−} spermatocytes relative to heterozygous

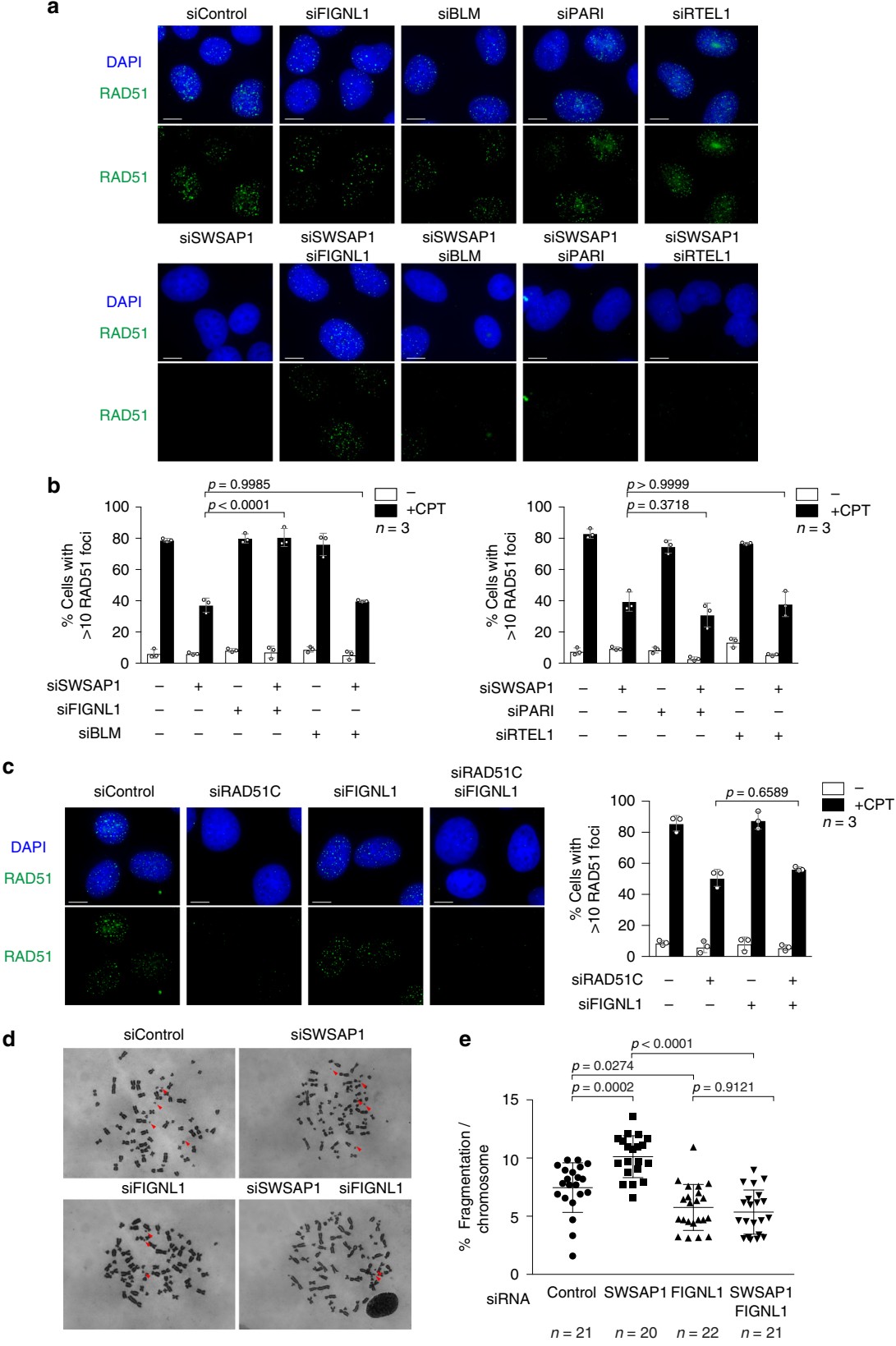

littermates ($47.1 \pm 3.3$ per a spread vs $133.8 \pm 3.7$, respectively) (Fig. 5f). We also examined the focus formation of a meiosis-specific RecA homolog, DMC1, and found the reduced DMC1-focus formation ($52.1 \pm 26.9$ per a spread in $Swsap1^{-/-}$ vs $120 \pm$

43.3 in its control; Fig. 5g). These results are consistent with the recent observation[33]. We observed relatively normal γH2AX-staining in $Swsap1^{-/-}$ leptotene cells (Supplementary Fig. 4e). These observations suggest that reduced RAD51- and DMC1-

**Fig. 3** FIGNL1 depletion suppresses RAD51-focus formation defect in SWSAP1-deficient cells. **a** Representative images of RAD51 immunofluorescence staining in siRNA-transfected U2OS cells. After 96-h transfection of various combinations of siRNA for anti-recombinases (FIGNL1, BLM, PARI, RTEL1) with siRNA with or without siRNA against SWSAP1, cells were treated with 100 nM of CPT for 22 h and immuno-stained for RAD51. Bar 10 μm. **b** Quantification of RAD51-positive cells in **a** was analyzed described in Fig. 1f. Data are mean ± s.d. (n = 3, three biological independent). Statistical significance was measured by Mann–Whitney's U-test. Statistics and reproducibility; see accompanying Source data. **c** Left; Representative images of RAD51 immunofluorescence staining in siRNA-transfected U2OS cells for FIGNL1 and or RAD51C. Right; Quantification of RAD51-positive cells in RAD51C- and FIGNL1-depleted cells was done as described in Fig. 1f. Bar 10 μm. Data are mean ± s.d.; Statistics and reproducibility, see accompanying Source data. **d** Representative images of metaphase chromosomes in siRNA-transfected cells. U2OS cells transfected with or without siRNA for SWSAP1 and or FIGNL1 for 96 h. Cells were arrested at metaphase by the treatment of colcemid for 2 h and chromosome spreads were prepared and Giemsa-stained. **e** Quantification of fragmented chromosomes in indicated siRNA-transfected cells. Percentages of fragmented chromosomes (shown by arrows) per a nuclear spread were calculated by dividing a number of fragmented chromosomes by a total number of chromosomes in a nucleus. More than 20 nuclei in each siRNA-treated cells were analyzed. Bar 10 μm. Data are mean ± s.d. Statistical significance was measured by Mann–Whitney's U-test. Statistics and reproducibility; see accompanying Source Data

focus formation during meiosis is due to inefficient RAD51/DMC1 assembly in response to damage rather than defective DSB formation in the absence of SWSAP1.

**Swsap1−/− cells are sensitive to DNA-damaging agents.** Since Swsap1-deficient mice show a defect in RAD51 assembly during meiosis, we assessed whether Swsap1 mutants exhibit HR deficiency during the mitotic cell cycle. Swsap1−/− immortalized fibroblasts were established and checked for DNA damage sensitivity. Swsap1−/− cells were highly sensitive to CPT, and also moderately sensitive to the DNA cross-linker, mitomycin C (MMC) (Fig. 6a, b). To assess the effect of Swsap1 deletion on RAD51 assembly in mitotic cells, we monitored RAD51-focus formation upon CPT-induced DNA damage. A reduction of ~25% in RAD51-foci positive cells was observed in Swsap1−/− fibroblasts relative to the wild-type control (53.4 ± 4.3% in wild-type vs 27.7 ± 5.8% in mutant), while γH2AX-focus formation was not affected in Swsap1−/− cells (Fig. 6c, d). These results further supported the hypothesis that Swsap1 is required for DNA damage-induced RAD51 assembly during mitosis.

## Discussion

In this study, we find an unique activity of the RAD51 paralogue, SWSAP1, to assist in RAD51 assembly. SWSAP1 protects RAD51 filaments by inhibiting a member of a novel class of RAD51 anti-recombinase, FIGNL1 (Fig. 7).

To be polymerized on the DNAs, RAD51 possesses two different interfaces to the adjacent RAD51 monomer/protomer. One is a short β sheet located between the N-terminal domain (NTD) and ATPase core domain, referred to as "β0", and the other is β5, a β sheet on the ATPase core domain in which one RAD51 monomer forms stable binding with β0 in the adjacent RAD51 monomer at 3′-end with a polarity. Interestingly, β0 contains the conserved Phe-X-X-Ala (FxxA) sequence and is structurally similar to the BRC motif of BRCA2, which directly binds to β5 of RAD51[7,30]. In this study, we showed that the FxxA motif of SWSAP1 is critical for RAD51-binding as well as RAD51-focus formation. This suggests that SWSAP1 uses the FxxA motif for binding to the β5 of RAD51. In addition, we revealed that a second conserved motif, PLQSMP, located downstream of FxxA motif is also important for the interaction with RAD51 (Fig. 1e). Consistent with previous observations of a second interaction motif in BRC repeats[34], downstream conserved amino acids could serve as interaction interface with a distinct pocket to RAD51.

If SWSAP1 binds to the RAD51 filament, it could be located in 3′-end of the filament which is constrained by the FxxA(β0)–β5 interaction (Fig. 7). A previous study has shown that the C. elegans RAD51 paralogue complex, RFS-1/RIP-1, binds to 5′-end of

the RAD51 filament and stabilizes the complex by preventing the dissociation of RAD51 from the end[35]. We envision that the RFS-1/RIP-1 complex utilizes a β5-like motif for the binding to β0 of RAD51 on 5′-end. Based on structural analysis of budding yeast Psy3-Csm2, which contains a dimer of two structural variants of Rad51/RecA fold, we also proposed that Csm2 of the Psy3-Csm2 dimer binds to 5′-end of Rad51-filament for the stabilization of the filament[11]. Unlike the homolog of the budding yeast Csm2, it is likely that, human SWSAP1, and thus the SWSAP1–SWS1 complex, binds to 3′-end of RAD51-ssDNA filaments in order to prevent the dissociation of RAD51 from that end.

Previous studies have identified FIGNL1 as a protein that binds to RAD51 and is involved in a step following RAD51 assembly in HR[29,36]. Here, we reveal a novel role of FIGNL1 as an anti-recombinase. We demonstrate that in vitro purified FIGNL1 promotes the dissociation of RAD51 from the RAD51–ssDNA complex. Although FIGNL1 belongs to AAA+ ATPase family, which has activity to remodel the conformation of different sets of proteins, we found that ATPase-deficient FIGNL1 has similar or slightly higher RAD51 dissociation activity than wild-type FIGNL1 (Fig. 4c and Supplementary Fig. 1i). These results indicate that ATPase activity of FIGNL1 is not critical for RAD51-dismantling activity. Given slightly higher activity of FIGNL1-KR than the wild-type, ATP hydrolysis of FIGNL1 may modulate its RAD51-dismantling activity. Based on the crystal structure analysis of human FIGNL1 dimer (PDB: 3D8B), the interface between human FIGNL1 monomers sandwiches ATP/ADP molecule, implying ATP-binding stabilizes dimer formation of FIGNL1 and facilitates its RAD51 dissociation activity.

Our observations suggest a novel mechanism by which FIGNL1 dismantles RAD51 from RAD51 filaments. We found that the RAD51-interaction domain (FxxA motif) of FIGNL1 is critical for this activity. Since RAD51 dissociation by FIGNL1 was eliminated by preventing ATP hydrolysis of RAD51, the binding of FIGNL1 to RAD51 may stimulate ATP-bound RAD51 to hydrolyze ATP to ADP, converting RAD51 to a low-affinity DNA-binding status. Taken together, we conclude that FIGNL1 is a new type of anti-recombinase, which disassembles RAD51 filaments from ssDNA in an ATPase activity-independent manner. Alternatively, FIGNL1, which binds to RAD51 in solution, may affect the equilibrium between association and dissociation states of RAD51 to the DNA (Fig. 7).

Whereas FIGNL1 disassembles RAD51 filaments from ssDNA, its activity is relatively low compared to other anti-recombinases[24–28]. This low RAD51-filament disruption activity of FIGNL1 could be explained by the absence of its N-terminal regions (1–284aa). The N-terminal region of FIGNL1 may enhance this activity. Furthermore, SPIDR and FLIP, FIGNL1-binding proteins, interact with the N-terminus of FIGNL1[29,37,38] and could support the full activity of FIGNL1.

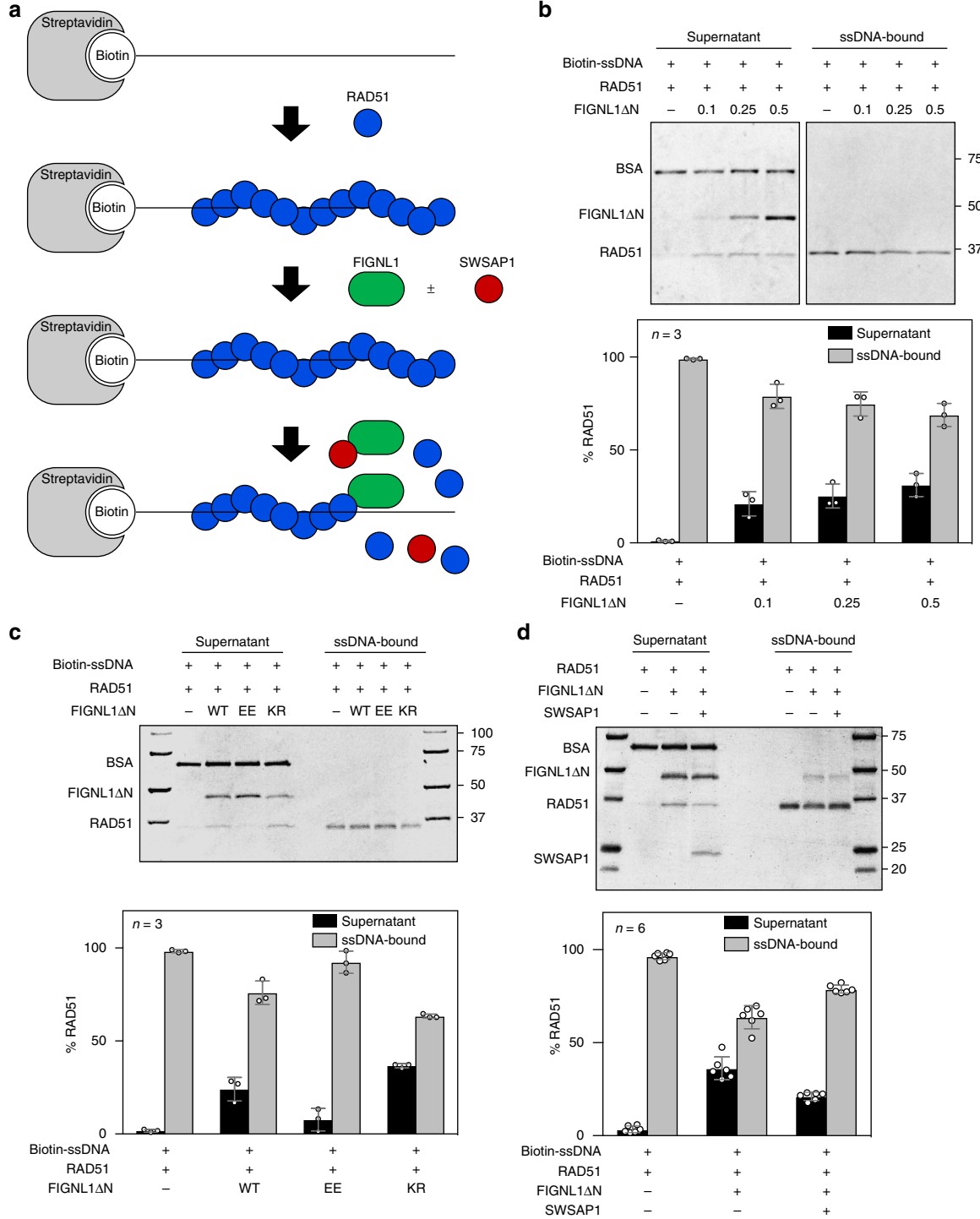

**Fig. 4** FIGNL1 dismantles RAD51 from ssDNA. **a** Schematics of RAD51 disassembly assay from ssDNA immobilized on magnetic beads. **b** RAD51 disassembly assay in the presence of FIGNL1ΔN. ssDNA-prebound to RAD51 was incubated with an increased concentrations of purified FIGNL1ΔN. After 30 min, supernatants and bound fractions were recovered. Top, a representative SDS-PAGE gel for supernatants and bound fractions stained with CBB. Bottom, Quantification of dissociated RAD51 (Supernatant) and ssDNA-bound RAD51. Intensity of each band of RAD51 was quantified by Imager. The values of RAD51 bands in the supernatant or ssDNA-bound fractions were divided by the total value of RAD51 bands (both in supernatant and bound fractions). Data are mean ± s.d. $n = 3$. Statistical significance was measured by two-tailed Student's $t$-test, see accompanying Source data. **c** RAD51 disassembly from ssDNA by FIGNL1 mutants. 0.2 μM of FIGNL1ΔN, FIGNL1ΔN-EE (F295E, A298E) or FIGNL1ΔN-KR (K447R) were added to ssDNA-pre-bound beads with RAD51. The binding of RAD51 was analyzed as described in **a**. Representative gel (top) and quantification (**b**) are shown. Data are mean ± s.d. $n = 3$. Statistical significance was measured by two-tailed student's $t$-test, see accompanying Source Data. **d** RAD51 disassembly from ssDNA in the presence of FIGNL1ΔN and or SWSAP1. 0.5 μM of SWSAP1 and 0.5 μM of FIGNL1ΔN were pre-incubated for 30 min and added to the RAD51-bound ssDNA beads. The binding of RAD51 was analyzed as described in **a**. Representative gel (top) and quantification (**b**) are shown. Data are mean ± s.d., $n = 6$. Statistical significance was measured by two-tailed student's $t$-test, see accompanying Source Data

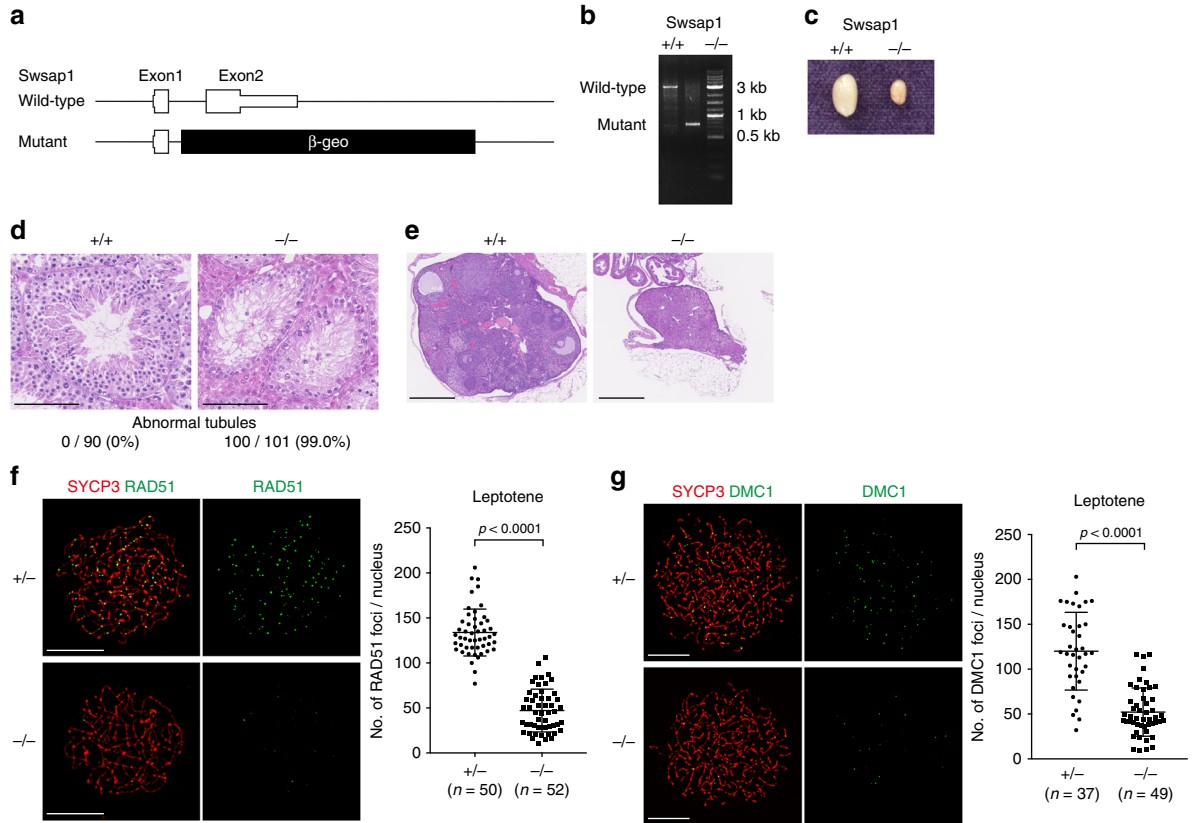

**Fig. 5** RAD51/DMC1-assembly defects during in *Swsap1* knockout (KO) mice. **a** Schematic of mouse *Swsap1* genomic locus. The genomic locus of mouse *Swsap1* shown to scale with KO construct (bottom). Exons are represented by boxes. **b** *Swsap1* PCR genotyping. Exon1-specific forward, *β-geo*-specific forward and reverse primers were used. Wild-type and mutant genes show 3 and 0.75 kb fragments, respectively. **c** *Swsap1* testis images. Representative images of *Swsap1*[+/+] and *Swsap1*[−/−] testis are shown. Bar 100 μm. **d** Testis sections of wild-type and *Swsap1* mutant mice. Cross sections of fixed testis were stained with HE. Top: Representative images of *Swsap1*[+/+] and *Swsap1*[−/−] seminiferous tubules are shown. Bottom, quantification of atrophic tubules is shown. Ninety tubules in *Swsap1*[+/+] and 101 tubules in *Swsap1*[−/−] were examined. **e** Cross sections of fixed ovary were stained with HE. Bar 500 μm. **f** RAD51 and SYCP3 immunofluorescence analysis of leptotene spermatocytes. Chromosome spreads were prepared and stained for RAD51 (green) and SYCP3 (red). Left, Representative images of *Swsap1*[+/−] and *Swsap1*[−/−] spermatocyte spreads are shown. Right, Quantification of RAD51 foci in leptotene spermatocytes. A number of RAD51 foci was counted per a nucleus. Data are mean ± s.d. Statistical significance was measured by Mann–Whitney's *U*-test. Statistics and reproducibility; see accompanying Source Data. **g** DMC1 and SYCP3 immunofluorescence analysis of leptotene spermatocytes. Chromosome spreads were prepared and stained for DMC1 (green) and SYCP3 (red). Left, Representative images of *Swsap1*[+/−] and *Swsap1*[−/−] spermatocyte spreads are shown. Right, Quantification of DMC1 foci in leptotene spermatocytes. A number of DMC1 foci was counted per a nucleus. Data are mean ± s.d. Statistical significance was measured by Mann–Whitney's *U*-test. Statistics and reproducibility; see accompanying Source Data

Previously, several proteins that dismantle RAD51 filaments have been identified. *S. cerevisiae* Srs2 DNA helicase is a primordial example of this class, which extends to BLM, RECQL5, and FBH1 helicases. Importantly, all these factors are DNA helicases that have DNA-binding activity and translocate on DNA molecules. It is shown that the ATPase activity of these helicases is critical for the dismantling activity of RAD51 filaments. Given that the ATPase motif of FIGNL1 is dispensable for its RAD51-filament disruption activity and major fraction of FIGNL1 does not bind to DNA, FIGNL1 belongs to a novel class of the RAD51 anti-recombinase since it does not work as an enzyme but may act in a stoichiometric manner.

A previous study showed that FIGNL1 depletion causes a defect in HR using DR-GFP system without affecting RAD51-focus formation[29], suggesting that FIGNL1 is involved in a later step of HR. We speculate that, similar to Srs2[39], FIGNL1 could optimize RAD51 filament on the DNA, which is suitable for efficient homology search and strand exchange etc. In other words, FIGNL1 may positively control the recombination by disrupting improper RAD51 ensembles.

We found that human FIGNL1 binds to human SWSAP1. In vitro, pre-incubation of SWSAP1 with FIGNL1 attenuates the RAD51-dismantling activity of FIGNL1, indicating that SWSAP1 antagonizes the anti-recombinase activity of FIGNL1. Consistent with this in vitro observation, the depletion of FIGNL1 in human cells suppresses a defect in the formation of DNA damage-induced RAD51 foci in SWSAP1-depleted cells. These data simply show that SWSAP1 protects RAD51 foci from FIGNL1. We propose that SWSAP1 is a novel type of human RAD51 paralogue, which regulates RAD51-filament formation by antagonizing the anti-recombinase, FIGNL1. Previously, the budding yeast Rad51 paralogue complex, Rad55–57, was shown to protect yeast Rad51 filaments from the Srs2 anti-recombinase[40]. Among the RAD51 paralogues, one class promotes RAD51 assembly by competing with the anti-recombinase. Interestingly, the protection function of SWSAP1 is very specific to FIGNL1 and is not observed against other anti-recombinases such as BLM, PARI, and RTEL1. Conversely, the other RAD51 paralogues such as RAD51C cannot antagonize the FIGNL1 activity. Previous observations that FIGNL1 does not bind to any RAD51

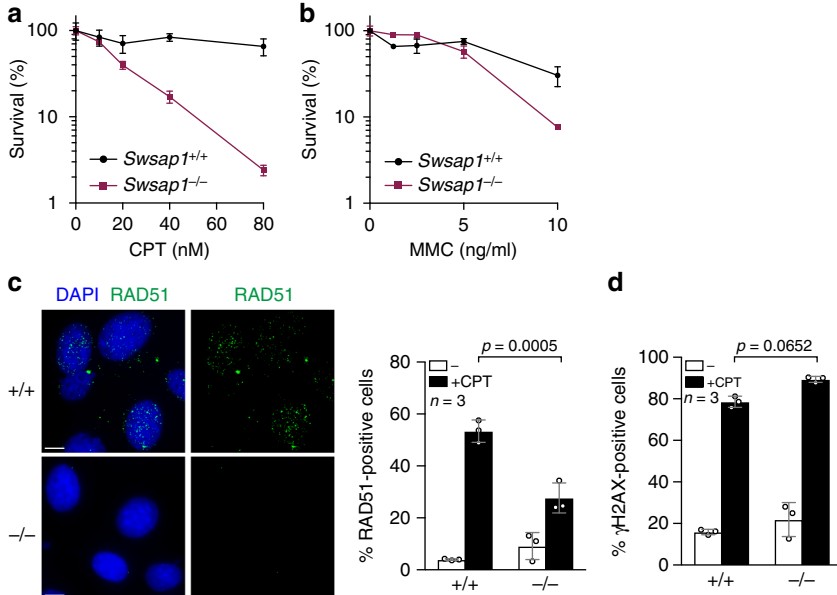

**Fig. 6** RAD51 assembly defects during mitosis in *Swsap1* knockout (KO) mice. **a**, **b** Clonogenic survival assays of immortalized *Swsap1*$^{+/+}$ and *SWSAP1*$^{-/-}$ cells exposed to different concentrations of camptothecin (CPT) and mitomycin C (MMC). A number of colonies with and without treatment of the DNA-damaging reagents were counted after 10–14 days culture. Data are mean ± s.e.m. Statistics and reproducibility, see accompanying Source data. **c** RAD51-positive cells in *Swsap1*$^{+/+}$ and *Swsap1*$^{-/-}$ cells after CPT treatment. *Swsap1*$^{+/+}$ and *Swsap1*$^{-/-}$ mouse embryonic fibroblast cells were treated with 100 nM CPT for 22 h. Cells were fixed and stained for RAD51 (green). Left, Representative images of RAD51-immunofluorescence staining. Bar 10 µm. Right, Quantification RAD51-positive cells in the indicated cell lines. More than 200 cells were counted. Data are mean ± s.d. Statistical significance was measured by Mann–Whitney's *U*-test. Statistics and reproducibility; see accompanying Source data. **d** Quantification of γH2AX-positive cells in *Swsap1*$^{+/+}$ and *Swsap1*$^{-/-}$ cells after CPT treatment. CPT-treated cells were stained for γH2AX. Data are mean ± s.d. Statistical significance was measured by Mann–Whitney's *U*-test. Statistics and reproducibility; see accompanying Source Data

paralogues suggests that SWSAP1 functions in an HR pathway distinct from BCDX2 and CX3 complexes[29]. In the future, we may expand the list of interactions between RAD51 mediators and anti-recombinases.

The dismantling activity of FIGNL1 requires the RAD51-binding motif, FxxA on its own and the RAD51 mediator activity of SWSAP1 needs the FxxA motif by itself. Simply, SWSAP1 competes for the β5 motif of RAD51 with FIGNL1. If SWSAP1 has a higher affinity to RAD51 than FIGNL1, this scenario is possible. However, we show that the interaction between SWSAP1 and FIGNL1 is also important for the anti-FIGNL1 activity of SWSAP1. Therefore, the association of SWSAP1 with FIGNL1 inhibits the activity of FIGNL1 either in solution or on the end of the RAD51 filaments. Further study is required to elucidate the mechanisms for how SWSAP1 protects RAD51 filaments from FIGNL1, particularly considering the SWSAP1-binding partner, SWS1, as well as the FIGNL1-binding proteins, SPIDR and FLIP.

Our results suggest that FIGNL1 inhibits inappropriate HR and SWSAP1 ensures HR when it is necessary. Alternatively, FIGNL1 may promote the fine-tuning of RAD51 assembly by working with SWSAP1. Our results, together with other studies, show that RAD51 filaments are highly dynamic and regulated by different combinations of positive and negative RAD51 factors, which may ensure the proper functional assembly on DNA substrates under different cellular conditions. Differential dynamics of RAD51 filaments may affect the fate of recombination intermediates into differential outcomes. In *Arabidopsis*, FIGNL1 has anti-crossover activity during meiosis, possibly by limiting RAD51-filament assembly[36,41]. In this case, FIGNL1 is not an anti-recombinase, but rather a pro-noncrossover recombinase. By balancing the activity of RAD51 mediators and remodelers under different

cellular contexts, a proper recombination pathway may be chosen by controlling RAD51-filament dynamics.

Unlike other RAD51 paralogue (*Rad51b*, *Rad51c*, *Rad51d* and *Xrcc2*) KO mice that are embryonic lethal[8,19–23], *Swsap1*$^{-/-}$ mice are viable, indicating that *Swsap1* is dispensable for embryonic development. Mild reduction of CPT-induced RAD51-focus formation in *Swsap1*$^{-/-}$ cells relative to the control could explain the dispensability of *Swsap1* for embryonic development. It suggests that, although *Swsap1* is critical for the stabilization of RAD51 filament in mitosis, there may be other factors to compensate for the absence of SWSAP1 in RAD51 assembly (Fig. 6c). Whereas embryonic development is unaffected in *Swsap1*$^{-/-}$ mice, 99.0% abnormal seminiferous tubules and lack of developing ovarian follicles in *Swsap1*$^{-/-}$ (Fig. 5d, e) imply that *Swsap1* is essential for meiosis and probably meiotic recombination. Indeed, *Swsap1*$^{-/-}$ mice showed a decreased number of RAD51 and DMC1 foci (Fig. 5f, g). These observations raise the possibility that *Swsap1* is a more specialized RAD51 paralogue for meiotic recombination than for mitotic HR. In addition, FIGNL1, a SWSAP1-binding protein, has been shown to limit crossover formation during meiosis in *Arabidopsis* and is highly expressed in mouse spermatocytes[36,41]. The formation of crossovers, which are essential for the proper segregation of homologous chromosomes, is highly regulated during meiosis. The SWSAP1–FIGNL1 axis is critical to ensure meiotic HR and, especially, CO formation.

## Methods
**Plasmid DNA and siRNA transfection**. For protein expression in human cells, 293T cells were transfected with indicated plasmid using XtremeGENE HP transfection reagent (Roche) according to the manufacturer's protocol. Five microgram of plasmid DNA and 15 µl of XtremeGENE HP were used for

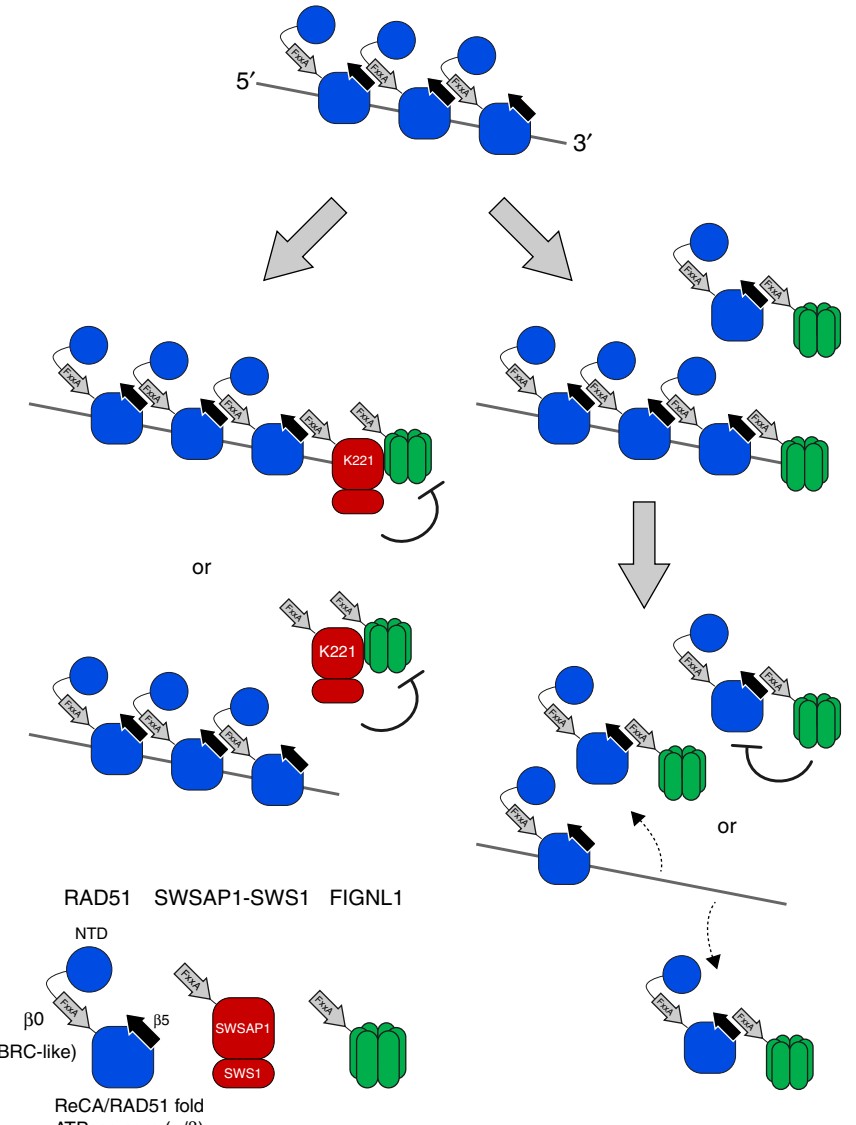

**Fig. 7** Model showing the regulation of RAD51 assembly by SWSAP1 and FIGNL1. (Right) FIGNL1 promotes the dissociation of RAD51 filament to prevent inappropriate HR. FIGNL1 binds to RAD51 through its conserved FxxA motif and facilitates RAD51 dissociation by inducing the conformational change of RAD51 or enhancing ATP hydrolysis of RAD51. (Left) SWSAP1 stabilizes RAD51 filament from anti-recombinase FIGNL1. SWSAP1 binds to RAD51 filament through its conserved FxxA motif. SWSAP1 interacts with FIGNL1 through C-terminus and inhibits FIGNL1's RAD51-filament disruption activity in the solution and/or at the end of ssDNA

293T cells (~$1 \times 10^6$ cells) on 6 cm-dish. Twenty-four hours after transfection, the media was replaced by DMEM (Gibco) containing 10% FBS. Seventy-two hours after transfection, cells were harvested by cell scraper and subjected to immunoprecipitation.

For siRNA transfection, U2OS cells were transfected with siRNA (Supplementary Table 1) using RNAiMAX transfection reagent (Invitrogen) according to the manufacturer's instruction. Four microliter of 50 μM duplex RNAs and 10 μl of RNAiMAX reagent were used for U2OS cells (~$0.7 \times 10^6$ cells) on 6 cm-dish. Seventy-two hours after siRNA transfection, cells were treated with 100 nM camptothecin (CPT; Wako) and subjected to immunofluorescence. Depletions of SWSAP1 and FIGNL1 were confirmed by depletion of FLAG-SWSAP1 and Myc-FIGNL1, respectively (Supplementary Fig. 2d, e). Protein expression of siRNA-resistant SWSAP1 mutants in siRNA-transfected cells were analyzed by western blot (Supplementary Fig. 2f, g).

**Immunoprecipitation (IP)**. After washing with PBS, cells were resuspended in 500 μl of benzonase buffer (20 mM Tris-HCl at pH 7.5, 40 mM NaCl, 2 mM MgCl₂, 0.5% NP-40, 50 U/ml benzonase [Millipore], 1× Protease inhibitor [Roche], 1× phosphatase inhibitor [Roche]), incubated for 10 min at 4 °C and cleared by centrifugation at $15,000 \times g$ for 10min[42]. The resultant extracts (WCE) were diluted with by adding 2 × WCE volume of No-salt IP buffer (25 mM Tris-HCl at pH 7.5, 1.5 mM DTT, 15% Glycerol, 1× Protease inhibitor, 1× phosphatase inhibitor).

Twelve microliter of EZview Red anti-Myc affinity Gel (Sigma) were added to 300 μl of diluted extracts and incubated for 2 h at 4 °C. After incubation, beads were washed three times with IP buffer (25 mM Tris-HCl at pH 7.5, 150 mM NaCl, 1.5 mM DTT, 10% Glycerol, 0.25% NP-40, 1× Protease inhibitor, 1× phosphatase inhibitor) and proteins were eluted with 50 μl of SDS sample buffer.

**Antibodies**. Anti-RAD51 (1:100 Santa Cruz, sc8364), anti-DMC1 (1:50 Santa Cruz, sc22768), anti-SYCP3 (1:500 abcam, ab97672), anti-γH2AX (1:500 Millipore, 05–636) antibodies were used for immunofluorescence staining of spermatocyte spread. Anti-RAD51 (1:500 Millipore, ABE257) and anti-γH2AX (1:1000 Millipore, 05–636) antibodies were used for immunofluorescence staining of U2OS cells. Anti-FLAG (1:3000 Wako 012-22384) and anti-Myc (1:3000 Nacalai 04362-34) antibodies were purchased from Wako and Nacalai, respectively. For pulldown assay, anti-FIGNL1 (1:500 abcam, ab173685) and anit-SWSAP1 (1:500 Thermo, PA5-25460) were used.

**Western blot**. IP and WCE samples were separated by 12% SDS-polyacrylamide gels and transferred onto a PVDF membrane (Millipore). The blots were blocked with 5% milk in TBST for 30 min and incubated with primary antibody overnight and secondary antibody for 30 min. Proteins were detected by alkaline phosphatase

kits (Nacalai). Uncropped images for each western blots are shown in Supplementary Fig. 6.

**Clonogenic survival**. SV40-immortalized mouse fibroblasts prepared as described below were seeded on 10 cm-dishes. After 8 h, cells were treated with indicated concentration of camptothecin (Wako) for 22 h and mitomycin C (Nacalai), continuously. Ten days after treatment, cells were stained in 4% crystal violet (Sigma). The number of colonies was counted.

**Immunofluorescence staining**. Cells were cultured on coverslips in the presence or absence of 100 nM camptothecin for 22 h. Cells were permeabilized with CSK buffer (10 mM PIPES at pH 6.8, 100 mM NaCl, 300 mM Sucrose, 3 mM MgCl$_2$, 1 mM EGTA, 0.5% Triton X-100, 1× protease inhibitor cocktail, 1× phosphatase inhibitor) for 5 min on ice and fixed with 2% PFA (Sigma) for 15 min at room temperature. The coverslips were blocked in TBST containing 3% BSA. Subsequently, coverslips were incubated with primary antibodies in TBST containing 3% BSA for overnight, washed 3 times with TBST and secondary antibodies for 1 h. After washing with TBST, the coverslips were mounted with Vectashield media (Vector Laboratories).

Stained spreads were observed using an epi-fluorescence microscope (BX51; Olympus) with a 60 × objective (NA1.3). Images were captured by CCD camera (CoolSNAP; Roper). Mouse spermatocyte spreads were observed using a computer-assisted fluorescence microscope system (DeltaVision; Applied Precision). The objective lens was an oil immersion lens (100×; NA, 1.35). Image deconvolution was performed using an image workstation (SoftWorks; Applied Precision), and afterwards processed using iVision (Sillicon), and Photoshop (Adobe) software tools.

**Spermatocyte spreads**. Testis was incubated in 2 ml testis isolation medium (104 mM NaCl, 45 mM KCl, 1.2 mM MgSO$_4$, 0.6 mM KH$_2$PO$_4$, 0.1% glucose, 6 mM sodium lactate, 1 mM sodium pyruvate) containing 2 mg-ml$^{-1}$ collagenase (Worthington) at 32 °C for 55 min[43]. Subsequently, testis was treated with 0.7 mg ml$^{-1}$ trypsin (Sigma) and 4 µg-ml$^{-1}$ DNaseI (Roche) at 32 °C for 15 min. The reaction was stopped by adding 20 mg-ml$^{-1}$ trypsin inhibitor in testis isolation medium (Sigma). The resultant suspension was filtered through a 70-µm cell strainer (Corning). After washing with testis isolation medium, cells were resuspended in 0.1 M sucrose and applied to a spot containing 1% PFA on glass slide. The slides were dried under moist condition for 2.5 h and dry condition for 1 h. After rinsing with DRIWEL (FujiFilm), slides were stored at −80 °C.

**Protein purification**. FIGNL1ΔN (N-terminal 284aa deletion) sequence was inserted into EcoRI site of pGEX-2T (GE Healtcare). FIGNL1ΔN-EE (F295E, A298E) and FIGNL1ΔN-KR (K447R) were generated by PCR-mediated site-directed mutagenesis. BL21 (DE3) was transformed with pGEX-2T-FIGNL1ΔN. Protein expression was induced by adding IPTG to final 0.2 mM for 3 h at 37 °C. Cell pellets were resuspended in buffer-A containing 50 mM Tris-HCl (pH 7.5), 150 mM NaCl, disrupted by sonication and cleared by centrifugation. Lysates were filtered and applied to GSTrap column (GE Healthcare) on an ÄKTA Pure system. Proteins were eluted with a gradient of glutathione. GST-FIGNL1ΔN containing fractions were incubated with thrombin (GE Healthcare) to remove GST tag at room temperature. After 16 h incubation, the fractions were applied to HiTrap SP (GE Healthcare). Proteins were eluted with a gradient of NaCl. FIGNL1ΔN-containing fractions were concentrated using Amicon Ultra 30 K centrifugal filter unit (Millipore). Purified proteins were analyzed by SDS-PAGE with Coomassie staining (Supplementary Fig. S3b, d). FIGNL1ΔN-EE (F295E, A298E) and FIGNL1ΔN-KR (K447R) proteins were purified as wild-type FIGNL1ΔN, and showed similar behaviors during the purification to the wild-type protein.

For pulldown assay, human SWSAP1 and SWS1 sequences were inserted into NdeI/SalI site of pET28a expression vector (Novagen). BL21 (DE3) codon plus strain was transformed with the pET28a–SWSAP1–SWS1 plasmid and cultured at 37 °C for the expression of His$_6$–SWSAP1–SWS1 complex. Protein expression was induced by adding IPTG to a final concentration of 0.2 mM for 16 h at 18 °C. Bacterial pellets were sonicated and cleared by centrifugation. Resultant bacterial lysates were applied to Ni-NTA (Qiagen). His-tagged SWSAP1 and SWS1 complex was eluted with a gradient of imidazole in buffer-A. Purified proteins were analyzed by SDS-PAGE with Coomassie staining.

For RAD51 disassembly assay, human SWSAP1 sequence was inserted into SmaI site of pGEX-2T vector, resulting in pGEX-2T–SWSAP1. BL21 (DE3) codon plus strain was transformed with the pGEX-2T–SWSAP1 plasmid. A colony was incubated in LB media at 37 °C overnight. The culture was transferred to large LB culture and incubated for 2 h at 37 °C. Protein expression was induced by addition IPTG to final 0.2 mM for 3 h. Bacterial pellets were resuspended in buffer-A and sonicated. Lysates were filtered and applied to GSTrap column on an ÄKTA Pure system. GST-SWSAP1 was eluted with glutathione containing buffer-A and treated with thrombin overnight to remove GST tag. After thrombin treatment, SWSAP1 protein was purified with HiTrap Q (GE Healthcare) with a gradient of NaCl and concentrated by Amicon Ultra 10 K centrifugal filter unit (Millipore). Protein concentration was determined by Bradford method (BioRad) using BSA as a standard.

**GST pulldown assay**. GST pulldown was performed using MagneGST pulldown system (Promega) according to the manufacturer's instruction. Extracts of bacteria expressing GST or GST–FIGNL1ΔN were prepared as shown in Protein purification and incubated with 40 µl GST beads for 1.5 h at 4 °C. Beads were washed with a buffer containing 50 mM Tris-HCl (pH 7.5) and 150 mM NaCl and incubated with 10 µl of 10 µM purified SWSAP1–SWS1 complex. After 1.5 h incubation, beads were washed three times with buffer containing 50 mM Tris-HCl (pH 7.5), 150 mM NaCl, 5% Glycerol, 10 mM DTT, and 0.5% Triton X-100. Bound proteins were eluted with 50 µl of 10 mM glutathione in the same buffer. Eluted samples and inputs were subjected to western blotting.

**RAD51 disassembly assay from ssDNA immobilized on magnetic beads**. 5′-biotinylated 83nt ssDNA was immobilized on Dynabeads MyOne Streptavidin (Invitrogen) in the presence of 0.5 mg-ml$^{-1}$ BSA and 0.015% NP-40. In Supplementary Fig. 5b, 5′-biotinylated 43nt, 83nt, and 153nt ssDNA were used. Beads were washed once and incubated with 0.4 µM RAD51 in 25 µl of the buffer containing 80 mM HEPES at pH 7.5, 20 mM Mg(OAc)$_2$, 16 mM ATP, 4 mM DTT, 100 mM NaCl, 0.04% NP-40 at 22 °C. After 20 min incubation, the beads were collected and incubated with purified FIGNL1ΔN, FIGNL1ΔN-EE or FIGNL1ΔN-KR in 20 µl of a buffer containing 20 mM HEPES at pH 7.5, 50 mM Mg(OAc)$_2$, 4 mM ATP, 1 mM DTT, 25 mM NaCl, 0.1% NP-40, 5% Glycerol, 15 µM φX174 Virion (NEB) for 20 min at 22 °C. In Supplementary Fig. 5a, 5 mM CaCl$_2$ was used instead of Mg(OAc)$_2$. Supernatants were reserved as unbound fractions. Beads were washed once with TE with 100 mM KCl, 0.01% NP-40 and the ssDNA-bound proteins were eluted with 20 µl SDS sample buffer. ssDNA-bound and unbound fractions were resolved on SuperSep Ace 12.5% gel (Wako) and stained with CBB or InstantBlue (Expedeon). Bands were quantified using ImageQuantTL (GE Healthcare). Percentages of RAD51 in the supernatant and ssDNA-bound fractions were calculated by dividing measured values of RAD51 bands in the supernatant or ssDNA-bound fractions by total values (Supernatant + ssDNA-bound).

**Measurement of ATPase activity**. Reactions were performed in buffer containing 25 mM Tris-HCl (pH 7.5), 100 ng µl$^{-1}$ BSA, 1 mM DTT, 10 mM MgCl$_2$, 2 mM ATP and 1 µCi [γ-$^{32}$P]ATP at 37 °C. After incubation, the reactions were stopped by the addition of 10 mM EDTA. The reaction products were separated by TLC PEI Cellulose (Millipore) and analyzed with a FLA9000 (GE Healthcare).

**Metaphase spreads**. Metaphase spread were performed according to a standard protocol[44,45]. Briefly, after ~90 h of siRNA transfection of U2OS cells, cells were treated with 0.2 µg ml$^{-1}$ colcemid (Gibco) for 2 h at 37 °C, washed with PBS and trypsinized. 5 ml of 75 mM KCl was added to cells resuspended in remaining PBS. After 20 min incubation, fixation solution (MeOH:AcOH; 3:1) was added dropwise. After 5 min incubation, cells were collected by centrifugation, washed with fixation solution and incubated in fixation solution overnight. The cell suspension was dropped on glass slide and air-dried. Slides were stained with 4% Giemsa solution (Merck) for 45 min and mounted with 50% glycerol. The chromosome image was captured using a microscope (BX51; Olympus, Japan) with a 100 × objective (NA1.3). Images were captured by CCD camera (SPOT Pursuit; SPOT imaging), and afterwards processed using SPOT image software (SPOT imaging), and Photoshop (Adobe) software tools.

**Animal care**. The care and use of mice in this study were performed in accordance with the guideline for proper conduct of animal experiments (Society Council of Japan). These procedures were approved by Institutional Animal Care committee in Institute for Protein Research, Osaka University (approval ID; 25-03-0).

**Animals**. Swsap1 targeting vector containing β-geo cassette, loxP site, exon2 and loxP site, was obtained from KOMP. The targeting vector was used to replace exon2 of mouse Swsap1 gene. The resultant ES cells were injected into ICR host blastocysts and implanted into pseudopregnant female mice. Obtained chimeric mice were bred to C57BL/6 mice. The heterozygous mice were mated with CAG-Cre mice to delete exon2. The resultant Swsap1$^{+/−}$ mice were bred to obtain Swsap1$^{−/−}$ mice. Genotypes were confirmed by PCR with the following primers: mSWSAP1-check-f(ATGGCGGAGGCGCTGAGGCGGGTGCTGAA), mSWSAP1-2step-f (CGGTTTCCATATGGGGATTGGTGGCGACGA), mSWSAP1-check-r2(TGCCTTATTTCCTGATCCAGGCTAGCTGTC).

**Histology**. For histology of testes, samples were fix in one ml of Bouin solution (Wako 023-17361) overnight and in 10% buffered formaldehyde overnight. For histology of ovaries, samples were fixed in one ml of 10% buffered formaldehyde overnight. After fixation, samples were embedded, sectioned and stained with hematoxylin–eosin. Stained samples were analyzed by NHSL (New Histo Science Laboratory).

**Cell line derivation**. Mouse ear fibroblasts were derived using standard protocol[46]. Briefly, small pieces of ear were treated with protease solution containing collagenase (Roche) and Dispase (Roche) for 45 min at 37 °C and incubated in DMEM containing 10% FBS, antibiotics and antimycotic overnight. Cells were filtered with

a 70-μm cell strainer and cultured for experiments. Cells were immortalized by large T-SV40 plasmid using XtremeGENE HP transfection reagent[44].

**Oligo-DNAs**. Sequence information of all primer DNAs used in the study is provided in Supplementary Table 2.

**Cell lines**. U2OS (ATCC HTB-96) and 293T (ATCC CRL-3216) were maintained in DMEM supplemented with 10% FBS and antibiotics.

**Statistical analysis**. GraphPad Prism 7 was used to assess statistical significance. For RAD51 foci number in leptotene spermatocytes, immunofluorescence and metaphase spread experiments of siRNA-transfected cells, Mann–Whitney's *U*-test and Tukey's multiple comparison test was used. All *p*-values are shown in Source file.

**Reporting summary**. Further information on experimental design is available in the Nature Research Reporting Summary linked to this article.

## Data availability

The source data in Figs. 1f, 2g, 3b, 3c, 3f, 4b, 4c, 4d, 5f, 5g, 6a, 6b, 6c, 6d, and Supplemental Figs. 1I, 3, 4a, 4b, 4c, 4e, 5a, 5b are provided as a Source Data file. All data are available from the corresponding authors upon requests.

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

## Acknowledgements

We thank Dr. A. Furukohri for discussion. We acknowledge Dr. Yuki Okada for instructing nuclear spreads for mouse testis, Dr. Ryo Sakasai for SWSAP1 cDNA, Dr. Junichi Miyazaki for CAG-Cre mice, and Dr. Hitoshi Kurumizaka for RAD51 expression plasmid and anti-DMC1 antibody. We are also indebted to members of the Shinohara lab, particularly Ms. H. Matsumoto for mouse care and Ms. C. Watanabe for cytological analysis of chromosomes. This work was supported by JSPS KAKENHI Grant Numbers 22125001, 22125002, 15H05973, and 16H04742 to A.S. and 17K17846 to K.M.

## Author contributions

K.M. and A.S. conceived and designed the experiments. K.M. performed the all experiments. S.K. carried out RAD51/DMC1 staining in testis. T.I. performed some of biochemical experiments. K.M. and A.S. analyzed the data, and prepared the manuscript.

## Additional information

**Competing interests:** The authors declare no competing interests.

