## [Peer Review File · Nature Communications]

Reviewers' comments:

Reviewer #1 (Remarks to the Author):

"Human RAD51 paralogue, SWSAP1, fosters RAD51 filament by regulating the anti-recombinase, FIGNL1 AAA+ ATPase" by Prof Shinohara and colleagues

FIGNL1 is a AAA+ ATPase that has been implicated in recombination through its interaction with RAD51 in work published by the Chen lab in 2013 (ref. 30). SWSAP1 is a structural RAD51 paralog, which is required for DNA damage-induced RAD51 focus formation. A recent paper (Nat. Comm. 2018) by the Jasin lab shows that SWSAP1 is required for meiotic recombination, but that the gene, unlike other RAD51 paralogs, is not essential for cellular life. This paper makes an important connection between SWSAP1 and FIGNL1 using cell biological and biochemical experiments showing structural and functional interaction.

The authors characterize the protein interactions between SWSAP1 and RAD51, as well as between SWSAP1 and FIGNL1. The antagonizing actions between SWSAP1 and FIGNL1 in RAD51 focus formation is well documented through cellular studies involving appropriate interaction mutants and back complementation experiments.

The biochemical dissection of the mechanistic details how FIGNL1 disassembles RAD51-ssDNA filament and how SWSAP1 inhibits this action shows that the disruption and rescue of RAD51 filament do not occur on ssDNA but through competitive protein binding in solution. This result might not be compatible with the assumption that both FIGNL1 and SWSAP1 are DNA-binding proteins and their effects should be on ssDNA lattice.

The *in vivo* work supports their conclusion that SWSAP1 is important for RAD51 filament formation, consistent with recent publication (Abreu et al. 2018 Nature Communications) that SWSAP1 is essential for mouse fertility but not for viability. It extends the published work by also analyzing somatic knockout cells.

Overall, the manuscript will make an important contribution to the field and identified an interdependent role of SWSAP1 and FIGNL1 in RAD51-dependent recombinational DNA repair.

Specific comments:

1) Titles of Figures 1, 2, 3, 5: I suggest replacing 'RAD51 assembly' with 'RAD51 focus formation', because that is what is measured. These changes should also be applied to the text, where the results are described. It is fine to interpret this defect in the discussion as a filament assembly defect, although it seems that the mechanism is more protection from disassembly.

2) Figure 4c, the top image shows similar level of ssDNA-bound RAD51 in the presence of WT and EE of FIGNL1 Δ N, or in the absence of any FIGNL1 Δ N. KR mutant works much better in inhibiting RAD51-DNA binding and the authors might need to provide an explanation to rule out the potential problems of inactive protein preps. An ATPase comparison of WT, EE, and KR might be helpful to show the activities of the purified proteins.

Why were the biochemical experiments conducted in the absence of Ca⁺⁺, which is known to be required to stabilize human RAD51 filaments? Under the Mg⁺⁺ conditions employed human RAD51 filaments are very brittle, as shown by the Wyman lab. It would be of interest to see, if FIGNL1 has an effect on filaments assembled under Ca⁺⁺ conditions to differentiate the two versions model.

3) FIGNL1 inhibits RAD51-ssDNA binding through a protein interaction in an ATP-hydrolysis-independent manner. However, the biochemical work involves FIGNL1 Δ N, instead of full-length

FIGNL1, which cannot link directly to the cellular work in Fig. 2g and 3. To confirm that FIGNL1 indeed inhibits RAD51 filament formation through competitive protein binding, cellular work should be done to test whether FIGNL1 KR complements as WT in cells treated with siFIGNL1 (Fig. 3b or 3e).

4) Since the mechanism of FIGNL1 seems to involve binding in solution, some of the interpretation and description should be adjusted to this. Figure 6 shows both modes of action.

5) How do the authors rationalize the conclusion by the Chen lab that FIGNL1 is required for HR (DR-GFP assay) with their model of anti-recombination? Interestingly, also the yeast Srs2 anti-recombinase is required for double-strand break repair by HR.

Reviewer #2 (Remarks to the Author):

RAD51 protein is a centerpiece in mammalian homologous recombination which is essential for maintaining genome integrity via faithful repair of both endo- and exogenous DNA damage. RAD51 binds to a ssDNA tail formed after double-stranded break processing, mediates search for homology within bulk of genomic DNA and performs strand exchange between homologous DNAs (sister chromatids in mitosis and, possibly, homologous chromosomes in meiosis). The initial step of homologous recombination – polymerization of Rad51 on ssDNA - is highly regulated by accessory proteins. While several RAD51 paralogs (RAD51B, RAD51C, RAD51D, XRCC2 and XRCC3) are implemented in facilitating formation and in stabilization of RAD51-ssDNA nucleoprotein filament, DNA helicases like FBH1, PARI BLM and RECQL5 are known to displace RAD51 from DNA.

The whole picture of up and down regulation and fine tuning of the RAD51 filament formation is far from being clearly defined, but the manuscript of Matsuzaki et al. makes a very important step towards its better understanding. The authors investigated the role of another human RAD51 paralogue, SWSAP1, and AAA+ ATPase, FIGNL1, in the regulation of t RAD51 filament homeostasis. They show that SWSAP1 and RAD51 physically interact and this interaction is important for RAD51 filament formation in vitro under condition of DNA damage. In addition to RAD51, SWSAP1 interacts with FIGNL1. Depletion of SWSAP1 in human U2OS cells results in RAD51 assembly defects that can be overcome by depletion of FIGNL1. Also, Matsuzaki et al. demonstrate that FIGNL1 dismantles ssDNA-RAD51 filament in vitro without the involvement of the ATPase activity of FIGNL1. Finally, they generated Swsap1 knockout mice and documented meiotic defect in testes of Swsap1^{-/-} males and sensitivity to DNA damaging agents in Swsap1^{-/-} mitotic cells. Based on these results, Matsuzaki et al. proposed a model in which SWSAP1 and FIGNL1 regulate RAD51 assembly acting in opposite directions, with FIGNL1 being a new type of down regulator capable of destabilization of the filament in the absence of ATPase activity.

The experiments described in this work are very carefully done, all the important controls are in place. Conclusions of the papers are well backed up by experimental data. The new findings are in a good correspondence with the previous literature. The manuscript is clearly written and easy to read. This reviewer would recommend this work for publication after the authors address a few points:

1. While it is clear that 57 N-terminal amino acids of SWSAP1 play a significant role in interactions with RAD51, more precise defining of the FxxA motif and especially the PLQSMF sequence as crucial for this interaction looks less convincing. Replacing any two of four amino acids with negatively charged or any six consecutive amino acids with alanines in any part of the protein may alter proper folding and, consequently, activity.
2. There is no discussion of stoichiometry of FIGNL1-RAD51 interaction. Additional experiments with filaments of different length might be needed to address this issue.
3. KR FIGNL1 mutant, deficient in ATP hydrolysis, clearly has better filament dismantling activity

than WT protein. It must be explained/discussed.

4. More detailed description of the phenotype of Swsap1 knockout mice needed, both in males and females.

5. Since Swsap1 knockout demonstrates a meiotic phenotype, at least, a discussion of the possibility that SWSAP1 regulates meiotic recombinase DMC1 is needed.

Responses to referees:

NCOMMS-18-28847

Our responses to referees are shown in green below and changes in the main text are shown in red.

Responses:

Reviewer #1 (Remarks to the Author):

1) Titles of Figures 1, 2, 3, 5: I suggest replacing 'RAD51 assembly' with 'RAD51 focus formation', because that is what is measured. These changes should also be applied to the text, where the results are described. It is fine to interpret this defect in the discussion as a filament assembly defect, although it seems that the mechanism is more protection from disassembly.

-Thanks. We replaced them in both titles and main text etc.

2) Figure 4c, the top image shows similar level of ssDNA-bound RAD51 in the presence of WT and EE of FIGNL1deltaN, or in the absence of any FIGNL1deltaN. KR mutant works much better in inhibiting RAD51-DNA binding and the authors might need to provide an explanation to rule out the potential problems of inactive protein preps. An ATPase comparison of WT, EE, and KR might be helpful to show the activities of the purified proteins.

-We measured an ATPase activity of wild type FIGNL1deltaN, FIGNL1deltaN-EE and FIGNL1deltaN-KR, which is now shown in Supplementary Figure 1g, h (page 12, last two sentence in the first paragraph). As expected, WT FIGNL1deltaN and -EE mutant proteins can hydrolyze ATP in a similar way. On the other hand, FIGNL1deltaN-KR mutant protein greatly reduced the activity.

In addition, we found that both FIGNL1deltaN KR and -EE mutant proteins exhibited similar behavior in purification steps, suggesting the similar folding of the all three proteins. This description was added in Methods (page 29, last sentence of the second paragraph).

Why were the biochemical experiments conducted in the absence of Ca⁺⁺, which is known to be required to stabilize human RAD51 filaments? Under the Mg⁺⁺ conditions employed human RAD51 filaments are very brittle, as shown by the Wyman lab. It would be of interest to see, if FIGNL1 has an effect on filaments assembled under Ca⁺⁺ conditions to differentiate the two versions model.

-The reason why we initially used Mg⁺⁺ is that we thought that ATPase activity or ATP-binding activity of FIGNL1 may be critical for the activity of the protein. Since dismantling activity of FIGNL1 seems to be independent of ATP hydrolysis, based on the reviewer's suggestion, we carried out *in vitro* disassembly assay of RAD51 filaments by FIGNL1 in the presence of Ca⁺⁺ (Supplementary Fig. 5a, page 11, second paragraph) and the results showed that, in the presence of Ca⁺⁺, FIGNL1

can not dissociate RAD51 from ssDNA. This suggests that ATPase activity of RAD51 is critical for the reaction. Alternatively, since RAD51 forms more stable filament in the presence of Ca^{++} than in the presence of Mg^{++} , FIGNL1deltaN activity to release RAD51 from the DNA is not strong enough to disrupt very stable Rad51-ssDNA (may need other factors or condition).

3) FIGNL1 inhibits RAD51-ssDNA binding through a protein interaction in an ATP-hydrolysis-independent manner. However, the biochemical work involves FIGNL1deltaN, instead of full-length FIGNL1, which cannot link directly to the cellular work in Fig. 2g and 3. To confirm that FIGNL1 indeed inhibits RAD51 filament formation through competitive protein binding, cellular work should be done to test whether FIGNL1 KR complements as WT in cells treated with siFIGNL1 (Fig. 3b or 3e).

-As suggested, we carried out the complementation assay by the FIGNL1-KR and -EE mutants and the results are shown in Supplementary Figure 1i (page 12, second paragraph). As shown, when both SWSAP1 and FIGNL1 are depleted together, DNA damage-induced Rad51 focus formation is observed as in the control (since FIGNL1 depletion suppresses RAD51-focus formation defect conferred by SWSAP1 depletion). When WT FIGNL1 (resistant to the siRNA) was introduced to the double depleted cells, the RAD51-focus formation was greatly reduced compared to the control. FIGNL1-KR mutant shows similar or slightly more reduction in RAD51 focus formation compared to the wild type. On the other hand, unlike wild-type FIGNL1 and -KR mutant, the introduction of FIGNL1-EE mutant did not reduce RAD51 focus formation. This result is consistent with the *in vitro* result of RAD51 disassembly assay.

4) Since the mechanism of FIGNL1 seems to involve binding in solution, some of the interpretation and description should be adjusted to this. Figure 6 shows both modes of action.

-As suggested, we added the model that FIGNL1 binds to free RAD51, which may change dynamics (equilibrium) of RAD51-ssDNA interaction in a “new” Figure 6 and explained the second possibility in the text (page 17, last sentence).

5) How do the authors rationalize the conclusion by the Chen lab that FIGNL1 is required for HR (DR-GFP assay) with their model of anti-recombination? Interestingly, also the yeast Srs2 anti-recombinase is required for double-strand break repair by HR.

-Given a clue by this reviewer, as yeast Srs2, we are thinking that dismantling activity of FIGNL1 may regulate the dissociation of RAD51 from recombination intermediates formed during normal HR, which might be relevant to “pro”-recombination activity of FIGNL1 found by the Chen lab. Alternatively, although not exclusive, the dynamic nature of RAD51, which is regulated by both positive and negative regulators (SWSAP1 and FIGNL1), might be a key to mediate

the homology search and strand exchange. We explained this in page 18, third paragraph.

Reviewer #2 (Remarks to the Author):

1. While it is clear that 57 N-terminal amino acids of SWSAP1 play a significant role in interactions with RAD51, more precise defining of the FxxA motif and especially the PLQSMP sequence as crucial for this interaction looks less convincing. Replacing any two of four amino acids with negatively charged or any six consecutive amino acids with alanines in any part of the protein may alter proper folding and, consequently, activity.

-We agree that, at this point, we can not rule out the possibility of altered protein folding of the protein. We added the possibility in the text (page 6, last sentence in the first paragraph).

2. There is no discussion of stoichiometry of FIGNL1-RAD51 interaction. Additional experiments with filaments of different length might be needed to address this issue.

-As suggested, we tried the RAD51-dismantling assay by FIGNL1 using a shorter and longer oligo-DNAs and showed the results in Extended data Figure 5b. We used 43mer, 83 mer and 153mer ssDNAs for the RAD51 dissociation and the results were shown in Supplementary Fig 5b. FIGNL1 could dissociate RAD51 from both 83mer and 153mer DNAs in a similar efficiency. On the other hand, RAD51 on 43mer DNA was a better substrate for FIGNL1 compared to two longer oligoDNAs. One of the interpretations is that RAD51 molecule on the end of the filament might be a good substrate for the FIGNL1 activity. This statement was added in page 11, the second half of the second paragraph.

3. KR FIGNL1 mutant, deficient in ATP hydrolysis, clearly has better filament dismantling activity than WT protein. It must be explained/discussed.

-As suggested, we discussed the possibility that FIGNL1 ATPase activity may indirectly regulate the RAD51-remodeling activity of FIGNL1. For example, ATP hydrolysis-dependent conformation change may alter the binding to RAD51, which is now described in the text (page 15, last two sentences in the second paragraph).

4. More detailed description of the phenotype of *Swsap1* knockout mice needed, both in males and females.

-As suggested, we carried out histological analysis of adult ovary and found that *Swsap1* KO female lack developing follicles compared to the control as shown in Figure 5e and Supplemental Fig 4d. Importantly, this is very much consistent with the phenotypes of *Swsap1* KO generated by Jasin group (Abreu et al., Nature Communications, 2018).

5. Since *Swsap1* knockout demonstrates a meiotic phenotype, at least, a discussion of the possibility that SWSAP1 regulates meiotic recombinase DMC1 is needed.

-As suggested, we discussed the role of SWSAP1 in DMC1 assembly for meiotic recombination. We tried immuno-staining analysis of DMC1 on the spreads of mouse spermatocytes and the KO spermatocyte spreads showed reduced DMC1 focus number (Figure 5g).

REVIEWERS' COMMENTS:

Reviewer #1 (Remarks to the Author):

"Human RAD51 paralogue, SWSAP1, fosters RAD51 filament by regulating the anti-recombinase, FIGNL1 AAA+ ATPase" by Prof Shinohara and colleagues

The revision improved further an already excellent study, which greatly advances our understanding of the role of FIGNL1 in recombination. The authors addressed in the revision all concerns and further strengthened the manuscript.

Regarding point 2 of reviewer #2 and the experiment added by the authors in response, I suggest adding an aspect in the interpretation of the results. Line 244ff, the authors describe the results using filaments assembled on different length oligonucleotide finding that the shortest substrate (43nt) was the best FIGNL1 substrate. The interpretation states "This suggests either end of RAD51 filament or the stability of RAD51 filaments affects the FIGNL1's RAD51 dismantling activity."

Maybe it could be clarified that RAD51 filaments dissociate protomers from their ends. Hence, this finding is very compatible with a solution-based sequestering model that is favored by other data.

Reviewer #2 (Remarks to the Author):

The authors have addressed all my concerns satisfactorily.

Response to reviewers:

Reviewer #1 (Remarks to the Author):

"Human RAD51 paralogue, SWSAP1, fosters RAD51 filament by regulating the anti-recombinase, FIGNL1 AAA+ ATPase" by Prof Shinohara and colleagues

The revision improved further an already excellent study, which greatly advances our understanding of the role of FIGNL1 in recombination. The authors addressed in the revision all concerns and further strengthened the manuscript.

Regarding point 2 of reviewer #2 and the experiment added by the authors in response, I suggest adding an aspect in the interpretation of the results. Line 244ff, the authors describe the results using filaments assembled on different length oligonucleotide finding that the shortest substrate (43nt) was the best FIGNL1 substrate. The interpretation states "This suggests either end of RAD51 filament or the stability of RAD51 filaments affects the FIGNL1's RAD51 dismantling activity." Maybe it could be clarified that RAD51 filaments dissociate protomers from their ends. Hence, this finding is very compatible with a solution-based sequestering model that is favored by other data.

-We replaced the sentence saying "FIGNL1 may dissociate RAD51 protomers from their ends. This finding is compatible with a solution-based sequestering model (see Discussion)" in last sentence of the second paragraph of page 11.